# WEE1 inhibitors synergise with mRNA translation defects via activation of the kinase GCN2

Jordan C. J. Wilson [1,2,3], JiaYi Zhu[4], Vanesa Vinciauskaite[5], Eloise G. Lloyd [1],
Simon Lam [1], Alexandra Hart[1], Chen Gang Goh[1], Fadia Bou-Dagher[1],
Hlib Razumkov[6,7], Lena Kobel [8], Zacharias Kontarakis [9], John Fielden[8],
Moritz F. Schlapansky[8], Joanna I. Loizou [2,3,10], Andreas Villunger [3,11],
Jacob E. Corn [8], Giulia Biffi [1], Glenn R. Masson [5], Stefan J. Marciniak [4],
Aldo S. Bader[1] & Stephen P. Jackson [1] ✉

Inhibitors of the protein kinase WEE1 have emerged as promising agents for cancer therapy. In this study, we uncover synergistic interactions between WEE1 small-molecule inhibitors and defects in mRNA translation, mediated by activation of the integrated stress response (ISR) through the kinase GCN2. Using a pooled CRISPRi screen, we identify GSPT1 and ALKBH8 as factors whose depletion confer hypersensitivity to the WEE1 inhibitor, AZD1775. We demonstrate that this synergy depends on ISR activation, which is induced by the off-target activity of WEE1 inhibitors. Furthermore, PROTAC-based WEE1 inhibitors and molecular glues show reduced or no ISR activation, suggesting potential strategies to minimise off-target toxicity. Our findings reveal that certain WEE1 inhibitors elicit dual toxicity via ISR activation and genotoxic stress, with ISR activation being independent of WEE1 itself or cell-cycle status. This dual mechanism highlights opportunities for combination therapies, such as pairing WEE1 inhibitors with agents targeting the mRNA translation machinery. This study also underscores the need for more precise WEE1 targeting strategies to mitigate off-target effects, with implications for optimising the therapeutic potential of WEE1 inhibitors.

WEE1 inhibition is attracting substantial interest as a target in cancer therapy, with several clinical trials using AZD1775, Zn-c3, or Debio0123 as a small molecule inhibitor of WEE1 (ClinicalTrials.gov: NCT03668340, NCT04439227, NCT05128825, NCT05743036, and NCT03968653). Furthermore, recently developed WEE1 inhibitors that include ACR-2316, IMP7068, and SY-4835 are due to undergo trials to test their safety and efficacy (ClinicalTrials.gov: NCT06667141, NCT04768868, and NCT05291182). The rationale behind the use of

[1]Cancer Research UK Cambridge Institute, University of Cambridge, Cambridge, UK. [2]Center for Cancer Research, Comprehensive Cancer Centre, Medical University of Vienna, Vienna, Austria. [3]CeMM Research Center for Molecular Medicine of the Austrian Academy of Sciences, Vienna, Austria. [4]Cambridge Institute for Medical Research (CIMR), University of Cambridge, Cambridge, UK. [5]Division of Cancer Research, School of Medicine, University of Dundee, Ninewells Hospital, Dundee, UK. [6]Department of Chemistry, Stanford School of Humanities and Sciences, Stanford University, Stanford, CA, USA. [7]Department of Chemical and Systems Biology, ChEM-H, Stanford School of Medicine, Stanford University, Stanford, CA, USA. [8]Department of Biology, Institute of Molecular Health Sciences, ETH Zurich, Zurich, Switzerland. [9]Genome Engineering and Measurement Lab, ETH Zurich, Zurich, Switzerland. [10]Breast Cancer Now Toby Robins Research Centre, The Institute of Cancer Research, London, UK. [11]Institute of Developmental Immunology, Biocenter, Medical University of Innsbruck, Innsbruck, Austria. ✉e-mail: Steve.Jackson@cruk.cam.ac.uk

WEE1 inhibitors is to both increase genotoxicity in S phase of the cell cycle and to override the cell cycle G2-M checkpoint through CDK1 and CDK2 overactivation.

WEE1 serves as an essential negative regulator of cyclin-dependent kinases CDK1 and CDK2 through tyrosine phosphorylation[1–5]. Inhibition of WEE1, resulting in overactivation of CDK1 and CDK2, can impact on several key cellular processes. These include impacting cell-cycle checkpoint control[6], replication origin activity[7], control over nucleotide pools via stabilisation of the ribonucleotide reductase subunit RRM2[8], and the protection of stalled replication forks[9].

To enhance the efficacy of WEE1 inhibitors and mitigate toxicity, previous studies have explored predictive biomarkers and combination strategies. WEE1 inhibitors have been noted to exhibit increased sensitivity in various genetic backgrounds, including loss-of-function TP53 with oncogenic KRAS mutations[10], deficiencies in SIRT[11], ATRX[12], RBM10[13], FBH1[14], as well as a reduction of the histone mark H3K36me3[8]. Despite attempts for patient stratification in the use of WEE1 inhibitors, their utility in the clinic has been limited due to several associated toxicities, including neutropenia, thrombocytopenia, nausea, and anaemia[15,16].

In this study, we uncover factors involved in mRNA translation that confer hypersensitivity to the WEE1 inhibitors. We find that this synergy is predicated on the ability of WEE1 small-molecule inhibitors to activate the integrated stress response (ISR) via the activity of the translation initiation factor eIF2α kinase, GCN2. The ISR is an evolutionarily conserved cellular signalling pathway that is initiated by the phosphorylation of the translation initiation factor eIF2α on Ser-51[17], leading to the attenuation of bulk protein synthesis and reprogramming of gene expression. This process is mediated by eIF2α kinases, including GCN2[18]. Our experiments show that several WEE1 small-molecule inhibitors activate the integrated stress response in various cancer and non-cancer cell lines via off-target ISR toxicity. Additionally, we note that WEE1 PROTAC degraders, which utilise WEE1 small-molecule inhibitors as warheads, elicit less ISR toxicity, while WEE1 molecular glues do not activate this pathway.

## Results

### CRISPR screen connects WEE1 inhibitors with mRNA translation defects

To identify novel genes associated with sensitivity or resistance to WEE1 inhibitors, we performed a pooled CRISPR interference (CRISPRi) screen in the untransformed, immortalised cell line RPE-1 inactivated for TP53 and expressing the transcriptional repressor dCas9-KRAB. This work used a DNA damage response (DDR) and cell cycle-focused library targeting over 2000 genes for transcriptional repression to identify CRISPR single guide RNAs (sgRNAs) that either increased or decreased relative cell viability in response to treatment with the WEE1 inhibitor, Adavosertib (AZD1775). The cells that received sgRNAs were treated with IC25 and IC95 AZD1775 doses over 9 days (Fig. 1a). Following successful quality control of the screen, which included confirming loss of representation of sgRNAs targeting essential genes (Supplementary Fig. 1b), subsequent DrugZ[19] bioinformatic analyses identified various factors predicted to contribute to WEE1 small-molecule inhibitor hypersensitivity or resistance. These include regulators of WEE1 stability, such as FAM122A[20]; factors downstream of WEE1, including CDK2 and CCNA2; components of nucleotide metabolism, RRM2 and DUT; and members of the anaphase-promoting complex, such as FZR1. We also noted that two genes involved in mRNA translation, GSPT1 and ALKBH8, emerged as hypersensitivity hits at the AZD1775 IC25 dose (Fig. 1b). GSPT1 is an essential factor involved in translation termination that facilitates release of newly translated peptides from the ribosome via its GTPase domain[21]. ALKBH8 is a tRNA modifier that can modify 5-carboxymethyl uridine at the wobble position of the tRNA anticodon loop[22,23].

The above findings suggested a connection between WEE1 inhibitors and the control of mRNA translation. To explore this relationship, we employed a 'molecular glue' compound that targets GSPT1 for degradation, CC-90009[24]. Thus, we found that CC-90009 was highly synergistic in combination with WEE1 inhibitors in both RPE-1 TP53−/− and HAP1 cell lines as assayed via crystal violet staining (Fig. 1c and Supplementary Figs. 2 and 3). Importantly, we found that three different small-molecule inhibitors of WEE1−AZD1775, Zn-c3, and Debio0123−exhibited strong synergy with CC-90009 in resazurin-based assays, which assess cell viability and metabolic activity. By calculating combined Loewe, Bliss, and HSA synergy consensus scores[25], we concluded that in RPE-1 TP53−/− cells, WEE1 inhibitors were considerably more synergistic in combination with CC-90009 than were small-molecule inhibitors that targeted the WEE1-related kinase PKMYT1 (RP-6306[26]), or the DDR and replication stress kinases ATR (AZD6738[27]) and CHEK1 (LY2603618[28]) (Fig. 1d and Supplementary Fig. 4).

Perturbations of GSPT1, including CC-90009 treatment, have been previously linked to ISR activation and the downstream accumulation of key transcription factors like ATF4[29,30]. Given this connection, we sought to investigate whether the synergy between WEE1 inhibitors and CC-90009 was dependent on ISR activation. A partial degradation of GSPT1 in combination with AZD1775 resulted in a substantial increase in ATF4 protein abundance (Fig. 1e). In parallel, we also observed a synergistic reduction in global mRNA translation, as indicated by a decrease in puromycin incorporation into nascent protein, which serves as a measure of global cellular protein synthesis. Cycloheximide treatment, used as a positive control for mRNA translation shutdown, resulted in the expected reduction of puromycin incorporation[31]. These findings suggested that the ISR is implicated in the WEE1i−CC-90009 synergy. Indeed, we found that the synergistic loss of cell viability from combination of WEE1 inhibitor and CC-90009 could be rescued by inhibiting the integrated stress response by ISRIB[32] or inhibiting GCN2 kinase activity by the compound GCN2iB[33] in RPE-1 TP53−/− and HAP1 cell lines (Fig. 1f and Supplementary Figs. 2 and 3). Furthermore, the loss of cell viability from treatments of AZD1775 or CC-90009 alone could also be rescued by ISRIB or GCN2iB compounds, which suggested that WEE1i−CC-90009 synergy arises from both drugs independently activating the ISR and GCN2 pathways (Supplementary Figs. 2 and 3).

### WEE1 small-molecule inhibitor treatment activates the ISR

Phosphorylation of the translation initiation factor eIF2α on Ser-51 to initiate the ISR is a vital signalling event that results in attenuating bulk protein synthesis as well as reprogramming gene expression. This is mediated by the action of eIF2a kinases, including GCN2. GCN1, another key player in this pathway, can facilitate the activation of GCN2 under stress conditions[34,35]. A dose escalation of AZD1775 in the RPE-1 TP53−/− cell line induced well-characterised markers of the ISR, including phosphorylation of GCN2 at Thr-899, a known autophosphorylation site that correlates with GCN2 activation[36,37], phosphorylation of eIF2α at Ser-51, and increased ATF4 protein levels. Inhibition of the GCN2 kinase by using 1 μM GCN2iB effectively blocked these markers of ISR induction (Fig. 2a). Accordingly, we found that depletion of either GCN2 or GCN1 by CRISPRi induced resistance to AZD1775 in resazurin assays (Fig. 2b, c). Furthermore, knockout of GCN1 in HEK293T cells significantly reduced the abundance of endogenous ATF4 protein induced by AZD1775 and, in a separate experiment, decreased the expression of a transfected ATF4 reporter compared to wild-type, GCN1-expressing HEK293T cells (Supplementary Fig. 5a−c). Demonstrating that these effects were not limited to the previously mentioned cell lines, we observed that AZD1775 treatment increased nuclear ATF4 protein abundance across a panel of cancer and non-cancer cell lines; with such increases being abrogated upon co-

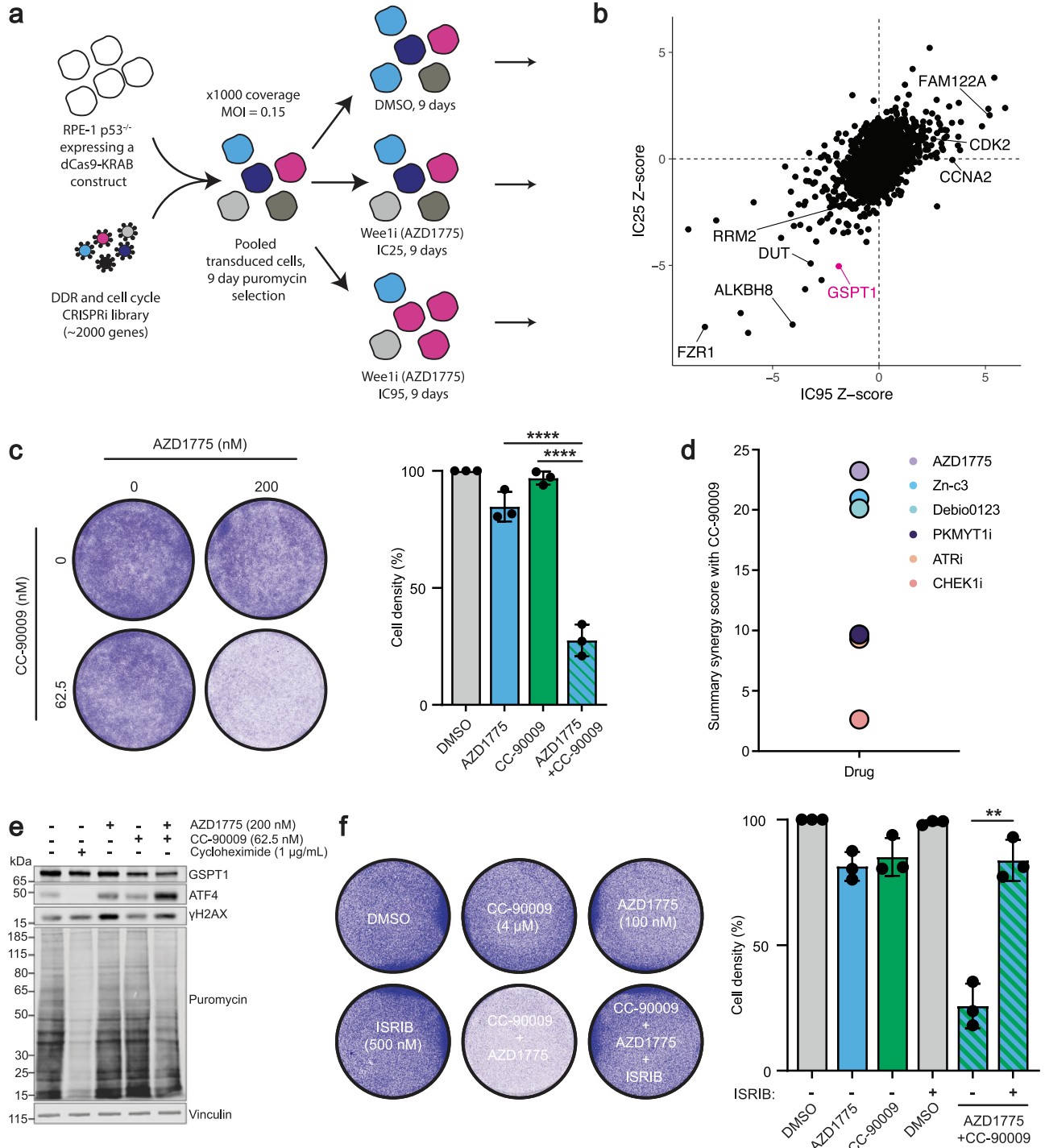

**Fig. 1 | A pooled CRISPRi screen reveals the perturbation of GSPT1 as a sensitivity hit for WEE1 inhibitors. a** Experimental design of pooled CRISPRi screen **b** CRISPRi screen results showing NormZ scores, calculated by DrugZ software, at IC95 (*x*-axis) and IC25 (*y*-axis) doses of AZD1775. Dosing information is available in Supplementary Fig. 1a. **c** Representative image of 200 nM AZD1775 and 62.5 nM CC-90009 compounds alone or in combination for 72 h in the RPE *TP53⁻/⁻* cell line in a 6 well plate format followed by quantification of the cell density (independant biological replicates *n* = 3). Bar charts are depicted with means ± SD, points represent each independant biological replicate. Statistical analysis was performed using one-way ANOVA with multiple comparisons: AZD1775 alone vs. Combination, *p* < 0.0001; CC-90009 alone vs. Combination, *p* < 0.0001. **d** Resazurin cell viability summary synergy scores (combined Loewe, Bliss and HSA synergy scores) of CC-90009 in combination with WEE1 inhibitors (AZD1775, Zn-c3, Debio0123), PKMYT1i (RP-6306), ATRi (AZD6738) or CHEK1i (LY2603618) in RPE *TP53⁻/⁻* cells in a 96 well plate format

(independant biological replicates *n* = 4). Heatmaps of the drug combinations are shown in Supplementary Fig. 4. **e** Western Blot showing 200 nM AZD1775 and 62.5 nM CC-90009 alone or in combination treated on the RPE *TP53⁻/⁻* cell line for 24 h. Cycloheximide (1 µg/mL) served as a positive control for global mRNA translation shutdown. Cells were treated with puromycin (5 µg/mL, 15 min) before harvesting. The probing of puromycin was run in parallel on a separate blot. A quantification of 3 independant biological repeats of this experiment can be found in Supplementary Fig. 17d–f. **f** Representative image of HAP1 cells treated with 100 nM AZD1775, 4 µM CC-90009, and 500 nM ISRIB compounds alone or in combination, for 72 h in a 6 well plate format, followed by quantification of the cell density (independant biological replicates *n* = 3). Bar charts are depicted with means ± SD points represent each independant biological replicate. Statistical analysis was performed using an unpaired two-tailed *t*-test (combination [AZD1775 + CC-90009] vs. Combination + ISRIB 500 nM), *p* = 0.0011. Source data are provided as a Source data file.

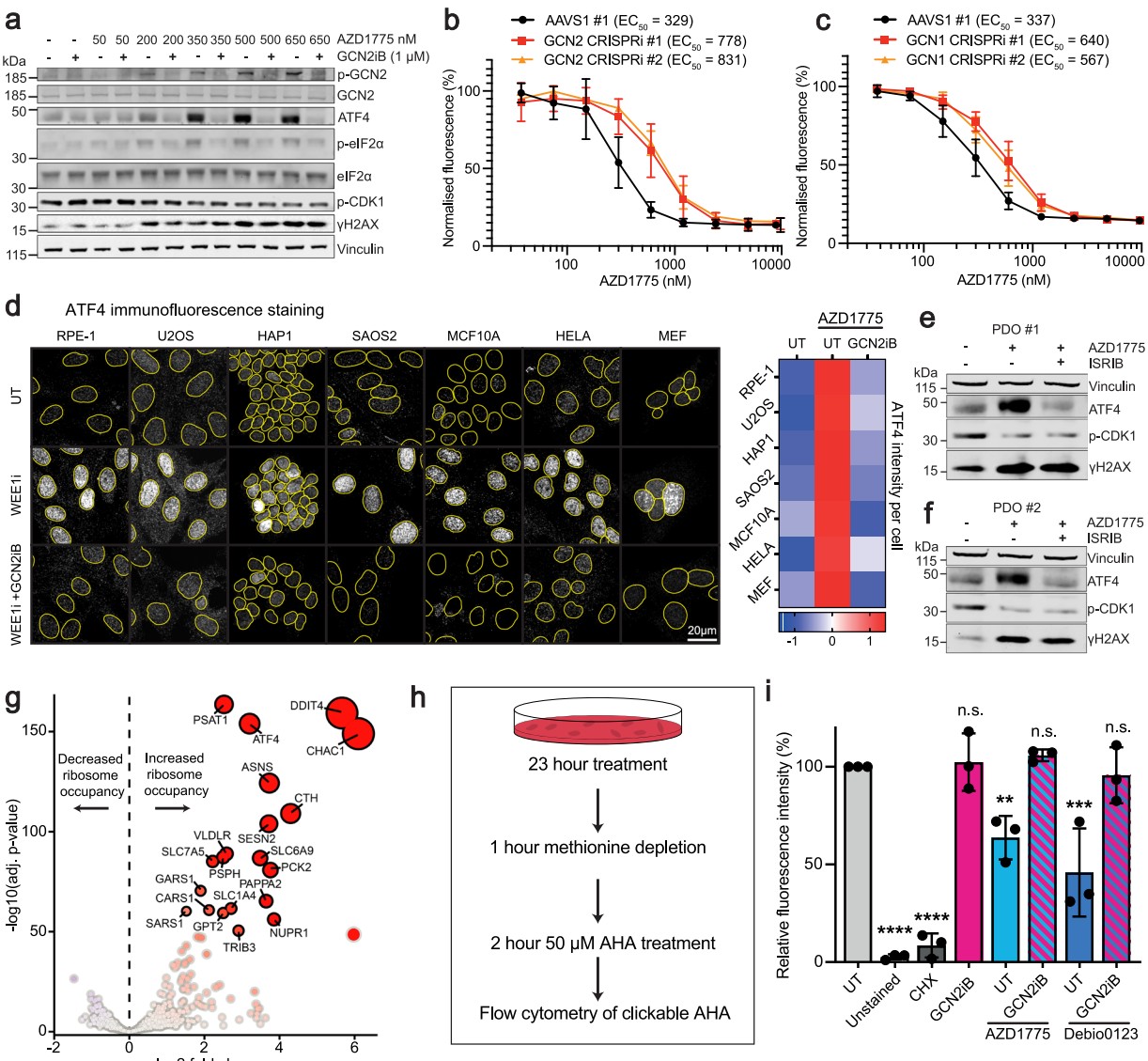

**Fig. 2 | WEE1 small-molecule inhibitor treatment activates the ISR. a** Western blot of RPE *TP53*⁻/⁻ cells treated for 24 h with increasing concentrations of AZD1775, with and without 1 µM GCN2iB. Bands of similar molecular weights were run in parallel on separate blots. Total protein (except for ATF4) served as loading controls. **b, c** Resazurin-based cell viability assay of RPE-1 *TP53*⁻/⁻ dCas9-KRAB cells expressing sgRNAs targeting GCN2, GCN1, or the AAVS1 locus. Cells were treated with varying concentrations of AZD1775 for 72 h in a 96 well plate format (independant biological replicates n = 4). Graphs are depicted with means ± SD. Validations of the CRISPRi knockdown of GCN2 and GCN1 are shown in Supplementary Fig. 9. **d** Representative immunofluorescence images of a panel of cell lines (RPE-1, U2OS, HAP1, SAOS2, MCF10A, HELA and MEF cell lines) probed for nuclear ATF4 treated with either DMSO, AZD1775 (650 nM for all cell lines except HAP1, which were treated with 300 nM), or AZD1775 in combination with 1 µM GCN2iB followed by row Z-score heatmap normalised per cell line summarising the immunofluorescence nuclear ATF4 intensity across different cell lines (independant biological replicates n = 4). **e, f** Western blots of two pancreatic ductal adenocarcinoma (PDAC) patient-derived organoids (PDOs) treated with either DMSO, 500 nM

AZD1775, or 500 nM AZD1775 + 500 nM ISRIB for 24 h. **g** Volcano plot of ribosome profiling data showing log2 fold change in ribosome occupancy in RPE-1 TP53⁻/⁻ cells treated with 650 nM AZD1775 versus DMSO for 10 h (n = 3 independant biological replicates). The x-axis denotes log2 fold change, and the y-axis represents −log10 of the adjusted two-sided p-value. **h** Summary of flow cytometry based AHA experiment on the RPE-1 *TP53*⁻/⁻ cell line. An example of the flow cytometry gating strategy is available in Supplementary Fig. 7a. **i** Bar chart of flow cytometry results showing the median fluorescence intensity of clickable AHA. RPE-1 *TP53*⁻/⁻ cells (6 well plate format) were treated with 1 µg/mL cycloheximide, 350 nM AZD1775, 1.5 µM Debio0123, and 1 µM GCN2iB alone or in combination. A cell population not treated with AHA served as an unstained negative control. Median fluorescence intensity results were normalised to the untreated control (UT) (independent biological replicates n = 3). Bar charts are depicted with means ± SD; points represent each independant biological replicate. Statistical analyses were performed by one-way ANOVA with multiple comparisons, comparing to untreated condition, ns = not significant, **p < 0.01, ***p < 0.001, ****p < 0.0001. Exact p-values are provided in the Source Data. Source data are provided as a Source data file.

treatment with 1 µM GCN2iB (Fig. 2d). Notably, we also observed increased ATF4 protein abundance in two pancreatic ductal adenocarcinoma (PDAC) patient derived organoids (PDOs) upon AZD1775 treatment (Fig. 2e, f). AZD1775 has previously been evaluated in pancreatic cancer clinical trials (NCT02037230, NCT02194829), highlighting its clinical relevance. Importantly, co-treatment with ISRIB

successfully rescued the ATF4 induction while preserving γH2AX levels.

Ribosome profiling, a technique that sequences RNA fragments that are protected by ribosomes[38], revealed that AZD1775 treatment reprogrammed mRNA translation, increasing ribosome occupancy on several ISR-related transcripts, including ATF4 and TRIB3, as well as

downstream effectors of ATF4, such as CHAC1 and PSAT1. Furthermore, regulators of the mTOR pathway, including DDIT4 and SLC7A5, also showed increased ribosome occupancy in response to AZD1775 treatment (Fig. 2g), coinciding with previous observations that link mTOR pathway perturbations to WEE1 inhibitor resistance[39–41].

To investigate whether the WEE1 inhibitors AZD1775 or Debio0123 affected global protein synthesis, we used L-AHA, a non-toxic methionine analogue that is incorporated into newly synthesised proteins in mammalian cells[42–44]. By measuring L-AHA incorporation, we assessed the rate of global nascent protein synthesis following treatment. We observed that both AZD1775 and Debio0123 reduced L-AHA incorporation, indicating a decrease in protein synthesis. Importantly, this reduction was rescued by co-treatment with 1 μM GCN2iB, suggesting that GCN2 activation is a key mediator of this effect (Fig. 2h, i). The ribosome profiling data revealed no significant changes in codon occupancy at the P or A sites of the ribosome following AZD1775 treatment, suggesting that the observed reduction in nascent protein synthesis was not due to specific codon stalling (Supplementary Fig. 6).

## WEE1 inhibitors synergise via ISR-dependent and -independent mechanisms

To determine which aspects of WEE1 inhibitor treatment are involved in ISR activation, we employed CRISPRi-based two-colour cell growth competition assays[45]. By utilising two distinct fluorescently labelled cell populations, this approach enabled us to evaluate the impact of different CRISPRi-mediated gene product depletions on cell fitness over time under WEE1 inhibitor treatment, both with and without ISR inhibition (Fig. 3a). GFP/mCherry fitness graphs were generated over 9-day treatments from the two-colour cell growth competition assays. The area under the curve (AUC) of the GFP/mCherry fitness graphs was compared across different drug treatments, including AZD1775, both alone or in combination with ISRIB, or GCN2iB (Fig. 3b).

CRISPRi-depleted cell populations that synergised with AZD1775, targeting gene products implicated in mRNA translation, nucleotide metabolism, and mitosis, were tested. The gene products tested were various components identified as hits in the CRISPRi screen as previously discussed, with the exception of PKMYT1, which was not present in the CRISPRi library but is a known synergistic interactor of WEE1 perturbations[46,47]. The relative loss of cell fitness upon WEE1 inhibitor treatment in GSPT1- and ALKBH8-depleted populations was rescued by co-treatment with either ISRIB or GCN2iB, suggesting that the synergy between WEE1 inhibitor and mRNA translation defects is ISR- and GCN2-dependent (Fig. 3c). By contrast, co-treatment with ISRIB or GCN2iB failed to significantly rescue the relative loss of cell fitness in DUT-, RRM2-, PKMYT1-, and FZR1-depleted populations upon WEE1 inhibitor treatment (Fig. 3d, e). These findings strongly suggest that the synergy between WEE1 inhibitors and defects in nucleotide metabolism and mitosis is independent of the ISR and GCN2 kinase activity.

A recent study has reported that WEE1 inhibitors exhibit strong synergy with the PKMYT1 inhibitor RP-6306[46]; a combination currently being evaluated for safety and efficacy in phase 1 clinical trials (ClinicalTrials.gov: NCT04855656). Using resazurin viability assays, we found that the synergy of AZD1775 and RP-6306 remained unaffected by co-treatment with ISRIB or GCN2iB (Fig. 3f). Given that WEE1 inhibitor and PKMYT1i synergy arises from the dysregulation of their shared phosphorylation target, CDK1, this suggests that CDK1 overactivation is not a driver of the WEE1 inhibitor-induced ISR phenotype. We also showed through immunofluorescence and western blotting analyses that depletion of other factors functioning downstream of WEE1, including CDK2, CCNE1, CCNE2, and CCNA2, did not impact AZD1775-induced ISR signalling (Supplementary Fig. 8).

Furthermore, the synergistic relationship between AZD1775 and hydroxyurea[8], which inhibits the ribonucleotide reductase enzyme[48] to deplete nucleotide pools, was not impacted by ISRIB or GCN2iB co-

treatments, as demonstrated by resazurin viability assays (Supplementary Fig. 12b). This further supported the notion that the synergistic interaction between WEE1 inhibitors and defects in nucleotide metabolism was independent of the ISR.

## WEE1 inhibitors activate the ISR independent of WEE1

To determine whether ISR activation following WEE1 inhibitor treatment was driven by inhibition of WEE1 itself or by off-target effects of the small-molecule compound, we used recently developed cereblon-dependent molecular glues that degrade WEE1 without requiring an ATP-competitive mechanism[49]. We established that in the RPE TP53$^{-/-}$ cell line, both 1 μM HRZ-1-057-1 and 1 μM HRZ-1-098-1 induced considerable degradation of WEE1 within 1 h (Fig. 4a). We next carried out sequential drug-addition studies, wherein after a 1-h molecular glue pre-treatment, cells were co-treated with DMSO or AZD1775 for 6 h, and ISR signalling was assessed (Fig. 4b). Notably, as revealed by western blotting, neither HRZ-1-057-1 nor HRZ-1-098-1 evoked ISR signals themselves; however, we continued to observe strong p-GCN2, p-eIF2α and ATF4 signals upon AZD1775 treatment, even after substantial degradation of WEE1 (Fig. 4c). A similar phenomenon was observed with the ATP-competitive small molecule WEE1 inhibitors Zn-c3 and Derbio0123, both currently in clinical trials, and the experimental compound WEE1-IN-4, which also activated the ISR despite significant WEE1 degradation (Supplementary Fig. 14a). From these data, we concluded that the ISR activation upon WEE1 inhibitor treatment was independent of WEE1 and due to off-target activity.

These observations raised the question of how the off-target activity was mediated. GCN2 has been recently reported to be activated by several ATP-competitive kinase inhibitors[36,50], prompting us to hypothesise that WEE1 inhibitors might activate the ISR via a similar mechanism. To test this, we conducted in vitro phosphorylation assays with GCN2 in the presence of ATP and various small-molecule inhibitors. Thus, we observed that phosphorylation of GCN2 induced by AZD1775 was comparable to that of the known GCN2 activator, and EGFR inhibitor, Neratinib[36]. The Debio0123 compound was less potent at inducing GCN2 phosphorylation compared to Neratinib and AZD1775, but was still considerably more active in this regard than the PKMYT1 inhibitor, RP-6306 (Fig. 4d, e and Supplementary Fig. 14b). Notably, a previous in vitro kinome profile study, which systematically evaluated the binding affinity of compounds across the kinome in vitro (403 wild-type and 65 mutant kinases), had shown that 0.5 μM AZD1775 was highly selective to the second domain of GCN2 along with WEE1, WEE2, PLK1, among others[51] (Supplementary Fig. 14c). The study validated dual WEE1-PLK1 inhibition by observing simultaneously attenuated levels of phosphorylated CDK1 (Tyr-15), a canonical WEE1 target[2], and TCTP (Ser-46), a previously described phosphorylation site of PLK1[52], in synchronised noncancer and cancer cell lines upon AZD1775 treatment. Taken together, these findings underscore that AZD1775 can modulate multiple kinase pathways in parallel.

To further test whether AZD1775 was capable of directly binding and activating GCN2, we used recombinantly expressed and purified human GCN2 in in vitro kinase assays and drug binding measurements. By monitoring ADP production of GCN2 in the presence of ATP and full-length human eIF2α substrate, we observed the characteristic 'bell shaped' curve of paradoxical activation. Activation of GCN2 was observed to peak at 36 nM AZD1775, with a drop to baseline activity at lower concentrations (EC$_{50}$ = 16 nM) and an inhibitory effect at higher concentrations (IC$_{50}$ = 89 nM) with a complete ablation of kinase activity at ~10 μM (Fig. 4f). Using thermal unfolding assays, we observed biphasic thermal behaviour of GCN2 with both negative and positive first derivative F350/F330 peaks (Fig. 4g). A biphasic profile was also observed with GCN2 in the presence of its physiological activator tRNA, showing a −2.1 °C shift in the negative peak (indicative of structural changes resulting in a reduction of tryptophan solvent

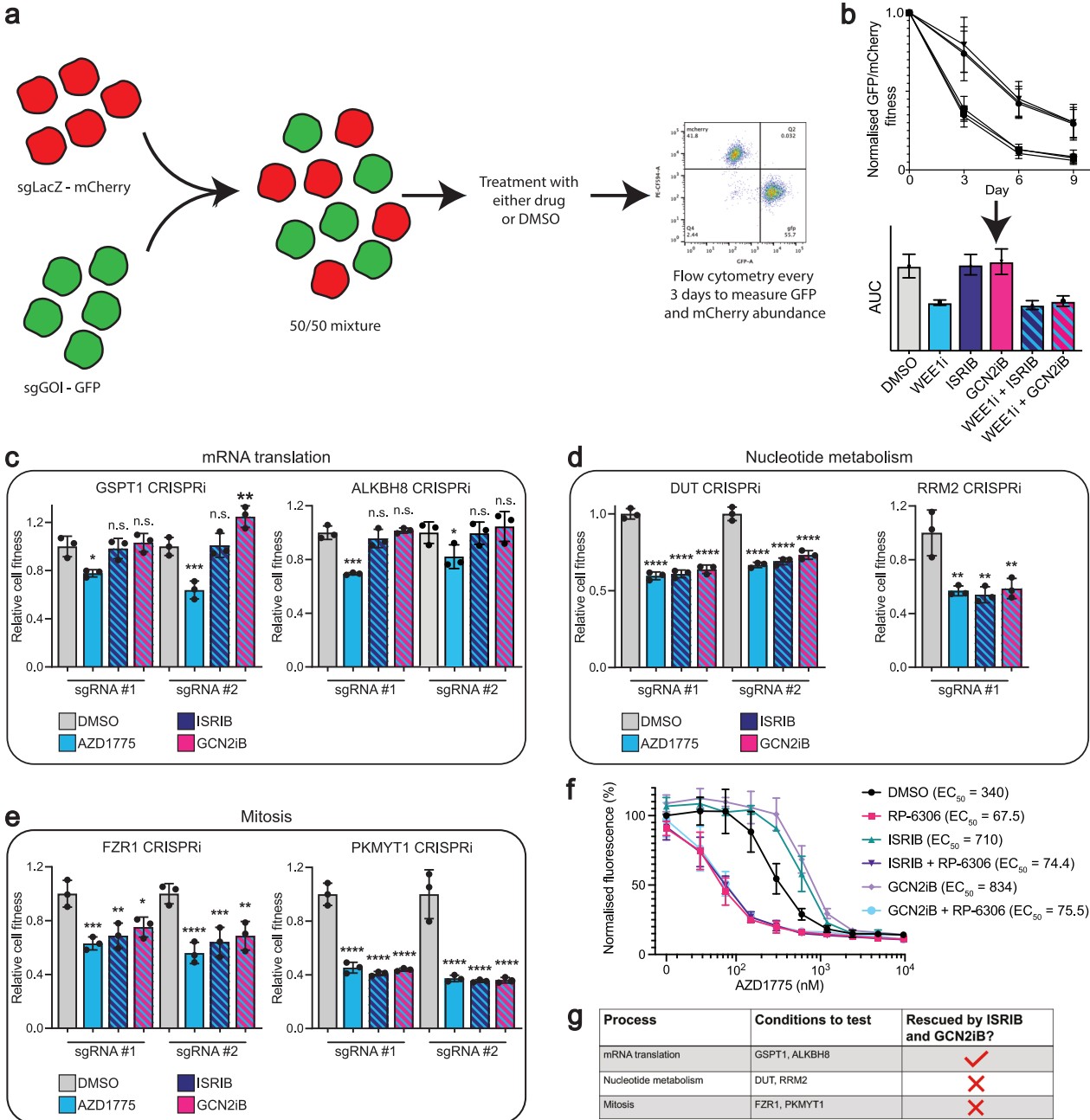

**Fig. 3 | WEE1 inhibitors synergise via ISR-dependant and ISR-independent mechanisms. a** Schematic showing flow cytometry based CRISPRi two-colour growth competition assays in RPE-1 *TP53*^−/− dCas9-KRAB cells. Cells were transduced with either sgLacZ-mCherry virus or sgGOI (gene of interest)-GFP. The mCherry and GFP expressing cell populations were mixed at a 50:50 ratio and treated with DMSO, AZD1775 (150 nM, 250 nM or 300 nM), 100 nM ISRIB or 1 µM GCN2iB, alone or in combination, for 9 days. Cells were assessed by flow cytometry, passaged and treated with fresh DMSO or drug every 3 days. The flow cytometry gating strategy is available in Supplementary Fig. 7b. **b** Schematic example of a normalised GFP/mCherry fitness graph. Values above 1 indicate the cell population expressing sgGOI causes increased relative cell fitness compared to cells expressing sgLacZ control, whereas values below 1 indicate decreased fitness. This is followed by an area under the curve that is generated from the GFP/mCherry fitness graph. **c**–**e** Bar charts quantifying the area under the curve of the normalised GFP/mCherry fitness graphs for different sgGOI-GFP populations vs sgLacZ-mCherry controls. Data were normalised to the respective DMSO-treated conditions. Values >1 indicate increased relative fitness compared to DMSO. Values <1 indicate decreased relative fitness compared to DMSO. Grey

bars represent DMSO only, blue bars represent AZD1775 treatment only, blue/navy striped bars represent AZD1775 + 100 nM ISRIB, blue/pink striped bars represent AZD1775 + 1 µM GCN2iB. AZD1775 concentrations used: 150 nM for GSPT1, ALKBH8, and PKMYT1 CRISPRi; 250 nM for RRM2 and FZR1 CRISPRi; 300 nM for DUT CRISPRi. Original GFP/mCherry fitness graphs for all conditions (including sgAAVS1-GFP vs sgLacZ-mCherry controls) as well as ISRIB and GCN2iB only treatment controls are provided in Supplementary Figs. 10–13, with CRISPRi knockdown validations in Supplementary Fig. 9. Bar charts are depicted with means ± SD, points represent each biological replicate (independant biological replicates *n* = 3). Statistical analyses were performed by one-way ANOVA with multiple comparisons, comparing to DMSO treatment, ns = not significant, *$p < 0.05$, **$p < 0.01$, ***$p < 0.001$, ****$p < 0.0001$. Exact *p*-values are provided in the Source Data. **f** Resazurin-based cell viability assay in RPE *TP53*^−/− cells treated with AZD1775 (varying concentrations) for 72 h, with or without: DMSO, 1 µM PKMYT1i (RP-6306), 100 nM ISRIB, 100 nM ISRIB + 1 µM PKMYT1i, 1 µM GCN1iB and 1 µM GCN1iB + 1 µM PKMYT1i (independant biological replicates *n* = 3). Graphs are depicted with means ± SD. **g** Summary table of the different CRISPRi backgrounds tested. Source data are provided as a Source data file.

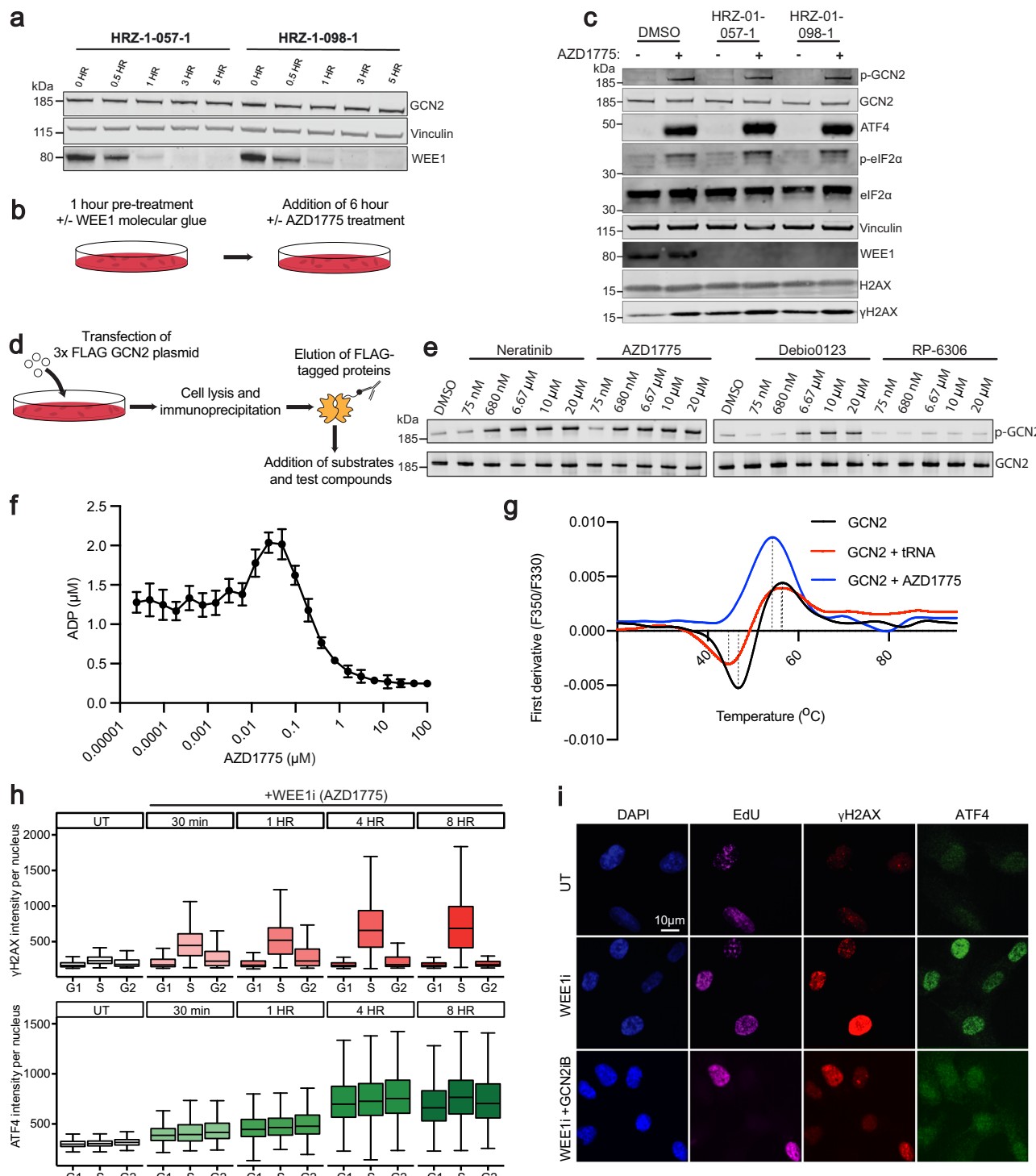

exposure) and only a subtle −0.2 °C shift in the positive thermal unfolding peak (indicative of exposed tryptophan residues). In contrast, AZD1775 addition eliminated the biphasic thermal profile of GCN2 and instead caused a more substantial −2.2 °C shift in the unfolding transition peak.

Additionally, we found that off-target ISR activation induced by AZD1775 was independent of cell-cycle status. Thus, immunofluorescent staining revealed no significant differences in ATF4 signals across the G1, S, and G2 phases of the cell cycle following AZD1775 treatment, whereas the DDR activation marker γH2AX (phosphorylated Ser-139 histone H2AX) was observed most strongly in S phase, as expected (Fig. 4h, i). Furthermore, we observe that both ATF4 and

γH2AX signals were induced very rapidly−within 30 min of AZD1775 treatment. Importantly, co-treatment with GCN2iB rescued ATF4 intensity but had no effect on γH2AX induction (Supplementary Fig. 15), further emphasising that the DNA damage and ISR activation induced by AZD1775 treatment are independent toxicities.

## WEE1 PROTAC elicits less ISR toxicity compared to AZD1775

Given that AZD1775 activates the ISR, we were interested in exploring how this compound would compare to its PROTAC derivatives that use AZD1775 as a targeting warhead to degrade WEE1 (Fig. 5a). Thus, we focused on ZNL-02-096, a previously developed WEE1 PROTAC that utilises AZD1775 as its warhead[49]. While AZD1775 exhibited strong

**Fig. 4 | WEE1 inhibitors activate the integrated stress response independent of WEE1 and independent of cell cycle status. a** Western blot showing the time course of WEE1 degradation in the RPE *TP53*[-/-] cell line following treatment with 1 µM HRZ-057-1 or HRZ-1-098-1. Data are representative of *n* = 2 independant biological replicates. **b** A schematic of the experiment in (**c**). **c** Western blot of the RPE *TP53*[-/-] cell line pre-treated with DMSO or 1 µM WEE1 molecular glues (HRZ-057-1 and HRZ-1-098-1) for 1 h (total treatment duration: 7 h), followed by DMSO or 650 nM AZD1775 for an additional 6 h. Bands of similar molecular weights were run in parallel on separate blots. Total protein (except for ATF4) served as loading controls. A quantification of three independant biological repeats of this experiment can be found in Supplementary Fig. 17a–c. **d** A schematic of the in vitro FLAG-tagged GCN2 experiment **e** Western blot of an in vitro experiment probing the total and phosphorylated GCN2 in the presence of DMSO, Neratinib, WEE1i (AZD1775 and Debio0123), and PKMYT1i (RP-6306). p-GCN2 and total GCN2 were run in

parallel on separate blots. A separate, independant biological experiment can be found in Supplementary Fig. 14b. **f** ADP-Glo assay of GCN2 mixed with a serial dilution of AZD1775 in the presence of ATP in a 384 well plate format (independant biological replicates *n* = 4). Graphs are depicted with means ± SD. **g** Thermal unfolding assays of GCN2 in the presence of 200 µM AZD1775 or 100 µM tRNA with a gradient of 0.5 °C/min from 25 °C to 90 °C. Dashed lines connect the negative and positive first derivative F350/F330 peaks to the *x*-axis. **h** Immunofluorescence analysis of nuclear ATF4 and γH2AX in the RPE *TP53*[-/-] cell line treated with 650 nM AZD1775 across multiple timepoints (independant biological replicates *n* = 3). Box plots show the median (centre line), the interquartile range (bounds of box), and the minimum and maximum values (whiskers). **i** Representative images showing ATF4 and γH2AX intensity in untreated (UT) cells and those treated with 650 nM AZD1775 alone or in combination with 1 µM GCN2iB for 24 h in the RPE *TP53*[-/-] cell line. Source data are provided as a Source data file.

synergy with CC-90009, ZNL-02-096 showed minimal synergy in RPE *TP53*[-/-] cells (Fig. 5b and Supplementary Fig. 16a). The depletion of GCN2 by CRISPRi gave rise to the resistance of AZD1775 as previously shown in Fig. 2b but not of ZNL-02-096 (Fig. 5c). To investigate this differential response, we performed further in vitro phosphorylation assays using recombinant GCN2 in the presence of ATP with either AZD1775 or ZNL-02-096 (Fig. 5d and Supplementary Fig. 16b). These assays revealed that ZNL-02-096 was less potent than AZD1775 at inducing GCN2 autophosphorylation when compared at equimolar concentrations.

The above data suggested that AZD1775 in its PROTAC form had less ISR mediated toxicity. Indeed, after 18 h of treatment of the RPE *TP53*[-/-] cell line, ZNL-02-096 induced less ATF4 protein abundance than AZD1775. However, despite eliciting a weaker ISR response, ZNL-02-096 induced greater γH2AX levels than an equivalent molar concentration of AZD1775 (Fig. 5e, f). Comparing across multiple timepoints, we observed that ZNL-02-096 treatment did give rise to ISR signals comparable to those elicited by AZD1775; however, with ZNL-02-096, these signals were transient and dissipated more rapidly over time. We also tested another WEE1 PROTAC, ZNL-02-047, which, like ZNL-02-096, employs AZD1775 as a warhead but differs in linker composition. Whereas ZNL-02-096 employs a hydrocarbon linker, ZNL-02-047 features a polyethylene glycol linker[53]. Notably, ZNL-02-047 induced considerably weaker ISR activation compared to both AZD1775 and ZNL-02-096 at the same 650 nM concentration (Supplementary Fig. 16c). Taken together, these findings suggested that PROTAC derivatives of AZD1775 may serve as a suitable alternative to AZD1775 if the desired outcome is to induce genotoxicity whilst limiting ISR activation.

## Discussion

We have investigated the determinants of sensitivity of RPE-1 *TP53*[-/-] cells to the ATP competitive WEE1 kinase inhibitor AZD1775. Our findings revealed that this WEE1 inhibitor synergised with depletion of mRNA translation factors GSPT1 and ALKBH8 through activation of the ISR and the GCN2 kinase. Furthermore, we established that treatment with AZD1775, as well as other WEE1 inhibitors—including Zn-c3, Debio0123, and WEE-IN-4—activated the ISR via GCN2. This aligns with a recent study reporting ISR activation following AZD1775 treatment in small-cell lung cancer (SCLC) cell lines[54]. Importantly, we have shown that this phenotype is conserved across cancer, non-cancer, human, and mouse cell lines as well as human patient-derived organoids; and additionally, that this WEE1 inhibitor-induced ISR activation is a "blanket" toxicity that is independent of cell-cycle status.

Prolonged ISR activation can lead to cell death[55], and it is well documented that DNA damage can also give rise to cell death[56]. Therefore, we suggest that currently exploited WEE1 inhibitors should be regarded as inhibitors that can elicit a dual toxicity—both ISR activation and genotoxicity. Through CRISPRi-based cell growth

competition assays, we have demonstrated that hypersensitivity to WEE1 inhibitors can be exploited via both ISR-dependent and ISR-independent mechanisms. Additionally, we also showed this with different drug combinations. Thus, we found that CC-90009, a GSPT1 degrader, exhibited strong synergy with WEE1 inhibitors through ISR activation, whereas the combination of WEE1 inhibitors with PKMYT1 inhibitors or hydroxyurea displayed potent synergy in an ISR-independent manner.

Notably, we found that recently developed WEE1 molecular glues that target WEE1 outside of the ATP-binding pocket showed no evidence of ISR activation but effectively induced on-target genotoxicity. By degrading WEE1 using molecular glues, followed by treatment with WEE1 small-molecule inhibitors, we demonstrated that these small-molecule inhibitors activated the ISR independent of the WEE1 kinase. Furthermore, our biochemical and in vitro data provided compelling evidence that WEE1 small-molecule inhibitors directly target the GCN2 kinase. Interestingly, while the WEE1 PROTACs ZNL-02-096 and ZNL-02-047 showed reduced capacity to activate the ISR overall, they did not induce detectable GCN2 degradation, suggesting that their diminished ISR activity result from altered binding kinetics rather than targeted protein degradation.

While WEE1 molecular glues and PROTACs represent valuable tools for mechanistic studies, these compounds were primarily developed for cell culture applications and their in vivo pharmacokinetic parameters remain unclear. In contrast, an immediate translational approach to minimise the ISR activation from WEE1 inhibitor treatment may be to combine WEE1 inhibitors with ISRIB, as ISRIB has more established in vivo toxicology profiles and has exhibited minimal overt toxicity at physiological concentrations where it demonstrated efficacy[32,57]. To assess the translational potential of this combination strategy, we tested WEE1 inhibitors combined with ISRIB in pancreatic patient-derived organoids, demonstrating that this approach can mitigate ISR activation whilst preserving DNA damage induction in clinically relevant models.

Recent publications have demonstrated that WEE1 inhibitors are not unique in their ability to activate the GCN2 kinase[36,50,58]. A plethora of seemingly specific kinase inhibitors can trigger GCN2 activation, highlighting a broader concern regarding the off-target engagement of this stress-response pathway. As such, activation of GCN2 may need to become a standard assay to evaluate the specificity of kinase inhibitors. We encourage future studies, and perhaps drug development pipelines, to incorporate GCN2 activation assays as part of a routine specificity screen.

Notably, depletion of both GCN1 and GCN2 rescued AZD1775-induced toxicity, and *GCN1*[-/-] cells showed greatly reduced ATF4 ISR signalling in response to AZD1775, low doses of GCN2iB and neratinib, but not to the PERK activator thapsigargin (Supplementary Fig. 5a–c; GCN2iB has been noted to activate GCN2 at low doses[58]). Furthermore, given that GCN1 depletion by CRISPRi conferred resistance to AZD1775 as shown in Fig. 2c, we tested whether this effect extended to other

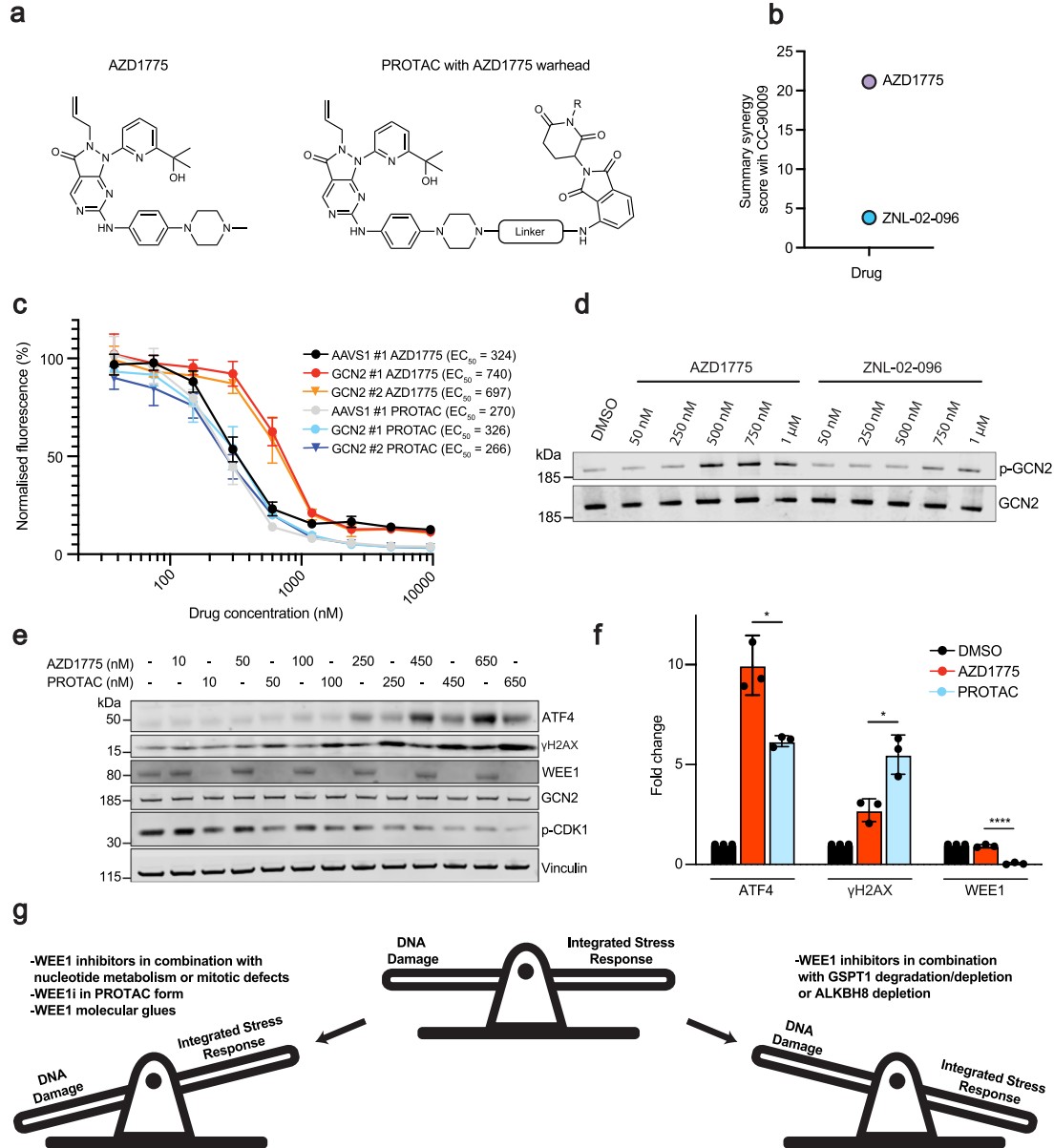

**Fig. 5 | WEE1 PROTAC elicits less ISR toxicity compared to AZD1775. a** Chemical structure of AZD1775 and PROTACs utilising AZD1775 as a warhead. The chemical structures were adapted on ChemDraw 25.0.2 from a previous publication[53]. **b** Resazurin cell viability summary synergy scores (combined Loewe, Bliss and HSA synergy scores) of CC-90009 in combination with AZD1775 or ZNL-02-096 in the RPE *TP53*[−/−] cell line in a 96 well plate format (independant biological replicates *n* = 3). Heatmaps of the drug combinations can be found in Supplementary Fig. 16a. **c** Resazurin cell viability assay with varying concentrations of AZD1775 or ZNL-02-096 treated for 72 h in a 96 well plate format in the RPE-1 *TP53*[−/−] dCas9-KRAB cell line expressing sgRNAs that target GCN2 or the AAVS1 locus (independant biological replicates *n* = 3). Graphs are depicted with means ± SD. **d** Western blot of an in vitro experiment probing the total and phosphorylated flag-tagged GCN2 in the presence of DMSO, AZD1775 and ZNL-02-096. p-GCN2 and total GCN2 were run in parallel on separate blots. Data are representative of *n* = 3 independant biological

treatments. We examined the impact of GCN1 depletion on sensitivity to neratinib and thapsigargin (Supplementary Fig. 5d, e). GCN1 depletion conferred resistance to neratinib but not to thapsigargin, which is consistent with our immunoblot findings. One possible explanation for the importance of GCN1 is that it 'primes' GCN2 for activation by certain small-molecule inhibitors, potentially by increasing ATP-binding pocket accessibility. This hypothesis may also

replicates. An additional independant biological experiment can be found in Supplementary Fig. 16b. **e** Western blot comparing AZD1775 and ZNL-02-096 18 h treatments in the RPE *TP53*[−/−] cell line. Data are representative of *n* = 3 independant biological replicates. **f** Quantifications of western blots of RPE *TP53*[−/−] treated with either DMSO, 650 nM AZD1775 or 650 nM ZNL-02-096 for 18 h (independant biological replicates *n* = 3). ATF4, γH2AX, and WEE1 were normalised to vinculin loading control. AZD1775 and ZNL-02-096 treatments were normalised to the vehicle to calculate the fold change of each condition. Graphs are depicted with means ± SD. Statistical analysis was performed using unpaired two-tailed *t*-tests comparing AZD1775 to ZNL-02-096: ATF4, *p* = 0.01246; γH2AX, *p* = 0.01331; WEE1, *p* = 0.00006. **g** Schematic showing the balance between the two independent toxicities of DNA damage and ISR activation for WEE1 inhibitor treatments. Source data are provided as a Source data file.

explain why WEE1 inhibitors synergise with mRNA translation defects, as these conditions could promote increased GCN1-GCN2 interactions.

In conclusion, we have established that various WEE1 small-molecule inhibitors activate the ISR via GCN1 and GCN2 in an off-target manner across a wide range of cell lines. We demonstrated that AZD1775 can synergise with various genetic backgrounds and drug treatments via both ISR-dependent and ISR-independent mechanisms

(Fig. 5g). Our study suggests that WEE1 small-molecule inhibitors used in the clinic lack precise specificity towards WEE1. However, using molecular glues and PROTACS that degrade WEE1, or combining existing WEE1 small-molecule inhibitors with ISRIB, causes greatly reduced ISR toxicity while maintaining WEE1-dependent genotoxicity. Therefore, such modalities may represent more precise and effective therapeutic alternatives for targeting WEE1-mediated toxicity without ISR related side-effects.

## Methods

### Cell culture

Cell lines used in this study were routinely tested for mycoplasma contamination, with no contaminations found. U2OS, SAOS2, HeLa, MEF and HEK293T cells were cultured in Dulbecco's modified Eagle's medium (DMEM, Gibco A4192101), hTERT RPE-1 (WT, $TP53^{-/-}$ and $TP53^{-/-}$ dCas9-KRAB) in Dulbecco's Modified Eagle Medium: Nutrient Mixture Ham's F-12 (DMEM/F-12, Gibco 11320033), HAP1 cells in Iscove's modified Dulbecco's medium (Sigma I3390-500mL), and MCF10A cells in mammary epithelial cell growth medium supplemented with bovine pituitary extract, human epidermal growth factor, insulin, hydrocortisone and cholera toxin (MEGM, Lonza CC-3150). All cell lines were grown at 37 °C and 5% $CO_2$. All cell lines except for MCF10A and RPE-1 (RPE-1 were cultured without additional L-Glutamine supplementation) were cultured in media supplemented with 10% foetal bovine serum (FBS), 2 mM L-Glutamine, 100 units/mL penicillin, and 100 µg/mL streptomycin. WT and $GCN1^{-/-}$ HEK293T cell lines were purchased from Abcam (ab255449 and ab266780).

hTERT RPE-1 $TP53^{-/-}$ dCas9-KRAB cell line was provided by the Jacob Corn laboratory and was generated by transduction with a lentiviral vector encoding the dCas9-KRAB and a Blasticidin resistance cassette with a low MOI. Cells were selected with Blasticidin, and single cell clones were seeded by cell sorting. Resulting clones were validated for CRISPRi activity by CD55 knockdown efficiency.

Methionine free medium was made as follows: 7.4 g of DMEM/F12 powder (D9785, Sigma), 0.077 g calcium chloride (Sigma), 0.046 g L-lysine (Sigma), 0.03 g L-leucine (Acros Organics), 0.0244 g magnesium sulphate (Fisher Scientific), 0.0306 g magnesium chloride (Sigma) and 0.6 g sodium bicarbonate (Sigma) were dissolved in 500 mL of MilliQ water and filtered using Millipore express PLUS 0.22 µm PES filter. Ten percent Dialysed FBS (Gibco) and 100 units/mL penicillin and 100 µg/mL streptomycin with additional 2 mM L-Glutamine supplementation were added.

Human organoids hM1a (annotated as PDO #1) and hF24 (annotated as PDO #2) have been previously published[59]. Human organoids were 3D cultured in Matrigel (354230, Corning) and complete media as described previously in ref. 60 at 37 °C with 5% $CO_2$. Human organoids were passaged twice weekly.

### Compounds

ATR inhibitor (AZD6738), CHEK1 inhibitor (LY2603618), WEE1 inhibitors (AZD1775, Zn-c3, and Debio0123), PKMYT1 inhibitor (RP-6306), ISRIB, Neratinib, and Thapsigargin were obtained from SelleckChem. GCN2iB and WEE1-IN-4 were obtained from Medchemexpress. WEE1 PROTAC (ZNL-02-096) was obtained from Tocris Bioscience. Hydroxyurea and cycloheximide were obtained from Sigma-Aldrich. WEE1 molecular glues (HRZ-1-057-1, HRZ-1-098-1) and WEE1 PROTAC (ZNL-02-047) were kindly provided by the Nathanael S. Grey laboratory. In the original publication, HRZ-1-057-1 is referred to as 'compound 1,' and HRZ-1-098-1 is referred to as 'compound 10'[49].

### Cell line generation for CRISPRi cell lines

To generate CRISPRi cell lines, lentiviruses were first generated in LentiX 293T cells by transfecting packaging plasmids psPAX2 (Addgene, #12260) and pMD2.G (Addgene, #12259) with the plasmid of interest using the transfection reagent TransIT-LT1 (Mirus Bio) according to the manufacturer's protocol. Seventy-two hours later, medium was collected, centrifuged at 2000g for 10 minutes, and the lentivirus-containing supernatant was stored at −80 °C. The RPE-1 $TP53^{-/-}$ dCas9-KRAB were cultured in 2 µg/mL puromycin 1 day post-transduction to select cells that had incorporated an sgRNA. All CRISPRi cell lines tested in the study were validated either by western blot or by RT-qPCR. sgRNA protospacers and primers used can be found in Supplementary Table 2.

### sgRNA cloning

CRISPRi sgRNAs were cloned into either mCherry- or GFP-containing lentiviral vectors (Addgene #185473 or Addgene #185474 plasmids). Forward and reverse primers for each sgRNA were annealed by pre-incubation at 37 °C for 30 min with T4 polynucleotide kinase (PNK; NEB), followed by incubation at 95 °C for 5 min and then ramp down to 25 °C at 5 °C/min. Annealed sgRNAs were ligated into the corresponding vector that had been digested with BsmBI restriction enzymes using T4 Ligase (NEB). All sgRNA protospacers are listed in Supplementary Table 2.

### CRISPRi library

For the DDR and cell cycle CRISPRi library design, the two strongest sgRNAs against each gene were selected from the CRISPRi-v2 library (PMID: 27661255). Where possible, this was determined using empirical scores; otherwise, the predicted activities from the CRISPRi-v2 library design algorithm were used. In case of multiple likely transcripts, sgRNAs against all transcripts in the CRISPRi-v2 library were included, such that some genes are targeted by 4 or more perfectly matched sgRNAs. sgRNA sequences of the library can be found in Supplementary Data 1. In addition to perfectly matched sgRNAs, the library also contained sgRNAs with single-base-pair mismatches relative to the intended target. For each perfectly matched sgRNA, four single-base-pair mismatched variants were included. These mismatched sgRNAs were prepared as a separate sublibrary, cloned independently, and then mixed with the perfectly matched sgRNA sublibrary after cloning. These mismatched sgRNAs were captured in CRISPR screen sequencing data but were not analysed or reported in this study. Protospacer sequences were appended with BlpI and BstXI restriction sites and PCR adapter. Oligonucleotides were synthesised by Agilent Technologies and cloned into the pLG1 library vector (pU6-sgRNA Ef1alpha-Puro-T2A-BFP). To ensure accurate representation of the library, Next-Generation Sequencing using the Illumina MiSeq platform was performed.

### CRISPRi screen

RPE-1-hTERT dCas9-KRAB $TP53^{-/-}$ cells were transduced with a lentiviral CRISPRi library (DDR CRISPRi allelic series library targeting 2294 genes with pLG1 (pCRISPRia-v2) backbone provided by the Corn Lab, ETH Zurich) at an MOI of 0.15 via spinfection. Two separate biological replicates of the screen were performed, i.e., two separate library transductions. The next day following transduction, fresh medium containing puromycin (2 µg/ml) was added. Cells were negatively selected with puromycin for a total of 9 days, at which point each replicate was divided into different treatments and subcultured every 3 days for a total of 9 days. Cell pellets were frozen at the end of the day-9 treatment (day 19 post-transduction), as well as day 3 post-transduction for gDNA isolation. Library coverage of at least 750 cells per sgRNA was maintained at every step for the DMSO and the IC25 arm, and at least 300 cells per sgRNA for the IC95 arm. gDNA from cell pellets was isolated using the QIAamp Blood Midi Kit (Qiagen, Cat# 51185), and genome-integrated sgRNA sequences were amplified by PCR using NEBNext Ultra II Q5 Mastermix (New England Biolabs, Cat# M5044L). Sample preparations were performed by amplifying the sgRNA with forward and reverse primers containing single TruSeq indexes. The final magnetic bead-purified products (purified using

SPRI MBSpure beads obtained from the Vienna Biocenter, Austria) were sequenced on an Illumina HiSeq3000 system to determine sgRNA representation in each sample. A 15% PhiX spike-in was used. To identify both synergistic and suppressor interactions, sgRNAs enriched or depleted in the treated samples were determined by comparison to control samples using DrugZ software[19]. Exorcise software was used to verify targeting of the sgRNAs to genome assembly GRCh38[61].

## Immunoblotting

Cells were collected in lysis buffer (50 mM Tris-HCl, pH 6.8, 1% SDS, 5 mM EDTA) and heated for 8 min at 95 °C. Protein concentrations were determined using a NanoDrop spectrophotometer (Thermo Scientific NanoDrop One) at 280 nm (with the exception of patient-derived organoid material). NuPage x4 LDS loading buffer (Invitrogen, NP0007) with 100 mM DTT was added to protein lysates at a final concentration of x1. SDS−PAGE was performed to resolve proteins on pre-cast NuPAGE Novex 4−12% Bis/Tris gradient gels (Invitrogen). Separated proteins were transferred to nitrocellulose (Sigma GE10600002), blocked for 1 h at room temperature in 5% BSA, 0.1% Tween 20 TBST, and immunoblotted with the indicated primary antibodies overnight at 4 °C. The nitrocellulose membranes were washed 3 times for 10-min washes in 0.1% TBST. The membranes were incubated in 1:10,000 secondary antibody for 1 h at room temperature and then washed 3 times for 10-min washes in 0.1% TBST. All secondary antibodies (except for probing ATF4) were incubated in fluorescent-conjugated secondary antibody (anti-rabbit secondary: LI-COR 926-68021, anti-mouse secondary: LI-COR: 926-32212). HRP-conjugated secondary antibody (Santa-Cruz, sc-2313) was used to probe for ATF4. Chemiluminescent signal was detected using SuperSignal West Pico PLUS (Thermo Scientific, 34580). Western blotting images were captured using ChemiDoc MP Imaging System (Bio-Rad) or LI-COR Odyssey M. A list of primary antibodies and dilutions used in this study can be found in Supplementary Table 1. Immunoblot images that were run in parallel on separate blots are indicated in their respective figure legends. Quantifications of Western blots were performed on Image Studio 6.0 and Fiji ImageJ 1.53t. software.

## Lysate generation and protein quantification of pancreatic cancer patient-derived organoids

Organoids were harvested in Cell Recovery Solution (354253; Corning) supplemented with protease inhibitors (11836170001; Roche) and a phosphatase inhibitor cocktail (4906837001; Roche), then incubated on ice for 30 min to dissolve the Matrigel. Cells were pelleted at 1500 rcf for 5 min, then lysed in 0.1% Triton X-100, 15 mmol/L NaCl, 0.5 mmol/L EDTA, 5 mmol/L Tris, pH 7.5, supplemented with protease and phosphatase inhibitors (11836170001, 4906837001; Roche) on ice for 30 min and vortexed. Protein-containing supernatant was obtained by centrifuging at $16,000 \times g$ for 10 min at 4 °C, from which the protein concentration was determined using DC protein assay (5000113-5; Bio-Rad).

## Crystal violet staining

Cells were added directly to drugs in a 6 well plate format with a total volume of 2 mL medium per well. The RPE $TP53^{-/-}$ cell line was seeded at 50,000 and 5000 cells per well for 3-day and 6-day treatments, respectively. The HAP1 cell line was seeded for 250,000 and 25,000 cells per well for 3-day and 6-day treatments, respectively. Cells were incubated with drug at 37 °C and 5% $CO_2$. Following 3 days or 6 days, the 6 well plates were washed with PBS and then stained with crystal violet (Pro-Lab PL.7002) for 30 min. Following this, the 6 well plates were washed with water, dried, and imaged on an Epson Perfection V800 scanner. The cell density was quantified using Fiji ImageJ 1.53t. software.

## Resazurin assays

Cells were added directly to drugs in a 96 well plate format with a total volume of 200 µL medium per well. RPE $TP53^{-/-}$, RPE-1 $TP53^{-/-}$ dCas9-KRAB, and HAP1 cell lines were seeded at 750, 1000, and 6000 cells per well, respectively. Cells were incubated with drug at 37 °C and 5% $CO_2$ for 72 h. Following this, pre-warmed resazurin reagent was added at a final concentration of 20 µg/mL and incubated for 3 h. Resazurin stock reagent was prepared by dissolving resazurin powder (Thermo Fisher Scientific, 418900250) in sterile PBS at 200 µg/mL and filtered with a 0.22 µm filter. After 3 h, the 96 well plates were read on the CLARIOstar plate reader with a 530−560 nm excitation wavelength and 590 nm emission wavelength. Two hundred microliters medium only outer wells were used as negative controls to account for background fluorescence. The fluorescence values were normalised to the DMSO-only-treated wells. Synergy scores from the normalised fluorescence intensities were calculated using the SynergyFinder software (synergyfinder.fimm.fi)[25].

## CRISPRi-based two colour competitive growth assay

RPE-1 $TP53^{-/-}$ dCas9-KRAB cells were infected at an MOI of -0.2 and treated with 2 µg/mL puromycin the next day and maintained in puromycin until the population was fully selected. This ensured that the transduced cell population all contained a puromycin resistance plasmid of NLS-mCherry LacZ-sgRNA or NLS-GFP GOI-sgRNA. Following selection, the mCherry- and GFP-expressing cells were mixed 1:1 (10,000 cells + 10,000 cells) and plated with or without drug in 12 well plate format. During the experiment, cells were subcultured and re-treated with or without drug every three days when they reached near-confluency. Cells were analysed on the A5 FACS Symphony (BD Biosciences), gating for GFP- and mCherry signal the day of the initial plating ($t=0$) and on days 3, 6, and 9. The efficiency of the knockdown by CRISPRi was analysed by western blotting or by RT-qPCR.

## Immunofluorescence

Cells were seeded in 24 well glass-bottom plates (Greiner 662892) at 50,000 cells per well and incubated for 24 h. Cells were treated with either WEE1 inhibitor or WEE1 inhibitor and GCN2 inhibitor for indicated times by exchanging medium with fresh medium containing drugs. Cells were washed once with PBS, fixed with 4% PFA for 10 min at room temperature, permeabilised with PBS + 0.1% Tween-20 (PBST) + 0.2% Triton X-100 for 20 min at room temperature, and blocked with 5% BSA in PBST for 1 h at room temperature. Antibodies were diluted to desired concentrations in 5% BSA PBST and incubated at 4 °C overnight. Cells were wash four times in PBST for 5 min at room temperature. Secondary antibodies tagged with Alexa Fluor 488 (Thermo, A11029) or Alexa Fluor 568 (Thermo, A11036) were diluted 1:1000 in 5% BSA PBST + 1 µg/mL DAPI and incubated for 1 h at room temperature. Cells were wash four times in PBST for 5 min at room temperature, and 2 mL PBS wash added to each well for storage until imaging. For cell cycle stratification, 1 µM EdU was added to cell culture medium 30 min before fixing, and following secondary antibody, a CLICK reaction was performed to conjugate Alexa Fluor 647 azide (Thermo, A10277) to the incorporated EdU as follows: 2 mM copper sulphate, 1 µM Alexa Fluor 647 azide, and 10 mM sodium ascorbate. Following CLICK, cells were wash twice with PBST for 5 min at room temperature before storing. Plates were imaged using an Opera Phenix Plus (Revvity) high content spinning disk confocal microscope, and analysis was completed in the Harmony software 5.2 (Revvity). Nuclei were identified through DAPI signal, and fluorescent intensities were quantified per nucleus as a mean intensity.

## AHA incorporation assay

RPE-1 $TP53^{-/-}$ cells were seeded in a 6 well plate with a total volume of 2 mL medium per well. Wells were treated with or without drug the next day for 23 h. Following this, the 6 wells were washed with PBS and

treated for 1 h in 2 mL of methionine free cell medium that contained the same drug concentrations as previous for each well. Fifty micromolar L-Azidohomoalanine/AHA (Thermo, C10102) was added for 2 h following a 1 h methionine depletion. The 6 well plates were washed with PBS, typsinised and harvested. Following fixation, click reactions using Click-iT Cell Reaction Buffer (Thermo, C10269) and 2.5 µM Alexa Fluor 647 Alkyne (Thermo, A10278) and subsequent washes were performed as per manufacturer instructions using the flow cytometry experimental protocol for adherent cells. Cells were analysed and gated on the A5 FACS Symphony (BD Biosciences) and analysed with FlowJo v.10.8.1. An example of the flow cytometry gating is shown in Supplementary Fig. 7a.

## RT-qPCR
RNA was extracted from snap-frozen cell pellets using the RNeasy mini kit (Qiagen) or Maxwell RSC simplyRNA Cells Kit (Promega). RNA extraction was as per manufacturer's instructions. During the extraction protocol, RNase-free plastic ware and solutions were used. After RNA extraction, isolated RNA was quantified using a Nanodrop spectrograph, and stored at −80 °C. cDNA was reverse transcribed from 1 µg of RNA using SuperScript IV VILO Master Mix (Invitrogen, 11756050), according to the manufacturer's instructions. cDNA was stored at −20 °C. qPCR was performed using 1 µL of cDNA, 10 µL of 2× Fast SYBR Green Master mix (Thermo Fisher Scientific, 4385612) and 500 nM forward and reverse primers, in a final volume of 20 µL. Primers were designed and ordered to span an exon-exon junction of the target genes (Sigma-Aldrich), and sequences are detailed in Supplementary Table 2. qPCR analysis was performed on a QuantStudio 5 Real-Time PCR System (Thermo Fisher Scientific), in technical triplicate. Gene expression changes were calculated using the $2^{-\Delta\Delta Ct}$ method[62]. RT qPCR results can be found in Supplementary Fig. 9b, c.

## Ribosome profiling
RPE $TP53^{-/-}$ cells were treated either with DMSO or 650 nM AZD1775 for 10 h in 15 cm² plates. Following the incubation, the growth media were aspirated and washed in ice-cold PBS/CHX (1X PBS supplemented with 100 µg/mL cycloheximide). Following this, 3 mL of ice-cold PBS/CHX was added per dish, cells were scraped extensively, and pelleted at $300 \times g$ centrifugation. Supernatant was aspirated, and the pellets were snap frozen. Flash frozen cell pellets were lysed in ice-cold polysome lysis buffer (20 mM Tris, pH 7.5, 150 mM NaCl, 5 mM MgCl₂, 1 mM DTT, 1% Triton X-100) supplemented with cycloheximide (100 µg/mL). For ribosome profiling (Ribo-seq), remaining lysates were digested in the presence of 35U RNase1 for 1 h at room temperature. Following RNA purification, PNK end repair, and size selection of ribosome-protected mRNA fragments on 15% urea PAGE gels, contaminating rRNA was depleted from samples using EIRNABio's custom biotinylated rRNA depletion oligos. Enriched fragments were converted into Illumina-compatible cDNA libraries. Ribo-seq libraries were sequenced on Illumina's Nova-seq 6000 platform with single-end sequencing to depths of 100 million raw read pairs per sample.

## Flag-tagged GCN2 in vitro assay
$EIF2AK4^{-/-}$ HeLa cells were transfected with pcDNA3.1/hygro(-)_hGCN2-3xFLAG[63] to express 3xFLAG-tagged GCN2. At 24 h post-transfection, cell lysates were prepared (50 mM Tris, pH 7.4, 150 mM NaCl, 1 mM EDTA, 1% Triton-X, protease and phosphatase inhibitors). FLAG-tagged protein was purified using anti-FLAG® M2 affinity gel slurry (Millipore, A2220) and eluted with 25 µg/ml 3xFLAG peptide solution (Pierce™ 3x DYKDDDDK, Thermo Scientific, A36805). Ten microliters of eluate were incubated with recombinant eIF2α-NTD, amino acids 2–187 (gifted by Heather Harding) to a final concentration of 1 µM and 500 µM ATP in reaction buffer: 50 mM HEPES, pH 7.4, 100 mM potassium acetate, 5 mM magnesium acetate, 250 µg/ml BSA, 10 mM magnesium chloride, 5 mM DTT, 5 mM β-glycerophosphate. Reactions were

incubated at 32 °C for 10 min, then immediately quenched with 5 µL of 6X SDS sample buffer for immunoblotting and heated for 8 min at 95 °C.

## GCN1 ATF4 reporter transfection
Wild-type and $GCN1^{-/-}$ HEK293T cells (ab266780) were transfected with pGL4.2_CMV_hATF4UTR:nLucSTOP to express the hATF4::nano-Luc reporter[63]. In 384-well plates, 103 cells in 20 µL per well were added to 5 µL of 5X treatment solution in 10% FBS-DMEM, yielding final concentrations of 5, 10, 35, or 70 nM GCN2iB, 3 mM histidinol, and 50, 100, or 200 nM AZD1775. After 6 h of incubation at 37 °C, 25 µL of NanoGlo® luciferase assay reagent was added to each well and bioluminescence (360–545 nm) acquired (Tecan Spark plate reader).

## GCN2 expression and purification (For ADP-GLO assay and thermal unfolding assay)
Expression and purification of recombinant human GCN2 were conducted as described previously in ref. 64. Human GCN2 (UniProt ID: Q9P2K8) was cloned into a baculoviral vector with an N-terminal twin StrepII tag followed by a TEV protease cleavage site. GCN2 was expressed in Sf9 cells grown at 27 °C for 55 h, then harvested via centrifugation, washed in ice-cold PBS, and snap-frozen in liquid nitrogen.

Cell pellets were thawed on ice with 100 mL of Lysis Buffer A (20 mM Tris, pH 8.0, 150 mM NaCl, 5% v/v glycerol, 2 mM β-mercaptoethanol (BME), and one cOmplete™ EDTA-free protease inhibitor tablet per 50 mL of buffer). Cells were lysed via probe sonication on ice for 5 min (10 s on/10 s off), followed by the addition of Benzonase (Millipore) at 2 U/mL. The lysate was then centrifuged at $140,000 \times g$ at 4 °C for 45 min. Protein purification was performed using an ÄKTA protein purification system (Cytiva). The supernatant was filtered through a 0.2 µm syringe filter before being loaded onto 2 × 5 mL StrepTrap HP Columns (GE Healthcare Life Sciences, 28-9075-47) equilibrated in Strep A Buffer (20 mM Tris, pH 8.0, 150 mM NaCl, 5% v/v glycerol, 2 mM BME) at a flow rate of 4 mL/min. The protein was eluted using a gradient of Strep B Buffer (20 mM Tris, pH 8.0, 150 mM NaCl, 5% v/v glycerol, 2 mM BME, 6 mM desthiobiotin). Peak fractions were analysed using SDS-PAGE to assess relative purity. GCN2-containing fractions were diluted using Q0 Buffer (20 mM Tris, pH 8.0, 5% v/v glycerol) to adjust the NaCl concentration to ~50 mM before being loaded onto a 5 mL HiTrap Q HP column (Cytiva, 17115401), equilibrated in QA Buffer (20 mM Tris, pH 8.0, 50 mM NaCl, 5% v/v glycerol, 2 mM BME) at a flow rate of 4 mL/min. The column was washed with 100 mL of QA Buffer, followed by the application of a gradient of QB Buffer (20 mM Tris, pH 8.0, 1 M NaCl, 5% v/v glycerol, 2 mM BME). GCN2-containing fractions were pooled and concentrated using a 50 mL Amicon centrifugation concentrator (50 kDa MWCO) to a volume of ~1 mL before being injected onto a HiLoad 16/60 Superdex 200 column (Cytiva/GE Healthcare, 28989335) equilibrated with GF Buffer (20 mM HEPES, pH 7.5, 150 mM NaCl, 2 mM tris(2-carboxyethyl) phosphine (TCEP)) at a flow rate of 1 mL/min. GCN2-containing fractions were concentrated using a 50 mL concentrator to a final concentration of ~3 mg/mL, aliquoted, and snap-frozen in liquid nitrogen.

## eIF2α expression and purification (For ADP-GLO assay)
Expression and purification of recombinant human eIF2α were conducted as described previously in ref. 64. DNA encoding full-length human eIF2α (NCBI reference number: NP_004085.1) was inserted into the vector pOPTH with an N-terminal His₆ tag followed by a TEV protease site. The plasmid was transformed into chemically competent BL21 Star (DE3) cells, which were grown overnight before being used to inoculate a 50 mL starter culture in 2×TY medium containing 0.1 mg/mL ampicillin. The starter culture was incubated at 37 °C for 90 min, then 10 mL of the starter culture was added to 4 × 900 mL of 2×TY medium containing ampicillin.

Cultures were incubated at 37 °C until the optical density (OD) reached 0.7, after which protein expression was induced by the addition of 0.3 mM isopropyl β-D-1-thiogalactopyranoside (IPTG). Cells were grown for an additional 3 h at 37 °C before being harvested, washed with ice-cold phosphate-buffered saline (PBS), and frozen in liquid nitrogen.

Bacterial cell pellets were lysed in 100 mL of Lysis Buffer (20 mM Tris-HCl, pH 8.0, 100 mM NaCl, 5% v/v glycerol, 2 mM β-mercaptoethanol (BME), 0.5 mg/mL lysozyme (Sigma L6876), 2 U/mL Benzonase, and one complete™ EDTA-free protease inhibitor tablet (Roche 04693132001) per 50 mL of buffer). Cells were lysed using a probe sonicator for 5 min (10 s on/10 s off) and then centrifuged at 140,000 × $g$ for 45 min at 4 °C.

The supernatant was filtered through a 0.2 μm syringe filter before being loaded onto a 5 mL HisTrap HP column (Cytiva 17524801) equilibrated in Ni A Buffer (20 mM Tris, pH 8.0, 100 mM NaCl, 5% v/v glycerol, 10 mM imidazole, pH 8.0, 2 mM BME). The protein was then eluted using a gradient of Ni B Buffer (20 mM Tris, pH 8.0, 100 mM NaCl, 5% v/v glycerol, 200 mM imidazole, pH 8.0, 2 mM BME). Protein purification then proceeded as described for GCN2. Proteins were concentrated to ~10 mg/mL and then snap-frozen in liquid nitrogen.

### tRNA purification (for thermal unfolding assay)

500 × $g$ of frozen beef liver was defrosted overnight at 4 °C. The liver was homogenised in 1 L of Lysis Buffer (20 mM Tris, pH 7.5, 1 mM DTT, 100 mM KCl). The homogenate was then centrifuged at ~5000–6000 × $g$ for 20–35 min at 4 °C. The collected supernatant was mixed 1:1 with water-saturated phenol and stirred for 30 min at room temperature (RT) in a fume hood.

The samples were then transferred into phenol/chloroform-resistant bottles (Nalgene) and centrifuged at ~5000–6000 × $g$ for 20–35 min at RT. The supernatant was diluted 1:1 with chloroform and stirred for 30 min at RT in a fume hood. The sample was centrifuged again as previously described, and the aqueous phase was collected while the supernatant and pellet were discarded. One-tenth of the aqueous phase volume of 3 M NaOAc (pH 5.0) was added to the aqueous phase and mixed. The sample was then diluted 1:2 with 95% ethanol, mixed, and stored at −20 °C overnight. The following day, the sample was centrifuged as previously described. The pellet was collected, left to air dry, and then resuspended in 100 mL of Lysis Buffer. The sample was centrifuged, and the supernatants were loaded onto an equilibrated 17 mL DEAE Sepharose gravity-flow column. The column was washed with 11 column volumes (CVs) of Lysis Buffer. The tRNA was eluted with 200 mL of Elution Buffer (20 mM Tris, pH 7.5, 1 mM DTT, 100 mM KCl, 1 M NaCl). The elution was mixed 1:2 with 95% ethanol and stored at −20 °C overnight. The sample was centrifuged at ~5000–6000 × $g$ for 35 min at 4 °C. The pellet was washed in 70% ethanol and left to air dry. The pellet was then resuspended in 2 mL of DEPC-treated water, and the concentration was measured. The sample was aliquoted and stored at −20 °C.

### Thermal unfolding assays

Thermal unfolding assays were conducted using a NanoTemper Panta using Prometheus Standard Capillaries (NanoTemper) (PR-C002). Briefly, GCN2 was defrosted and centrifuged at 20,000 × $g$ for 10 min, and then diluted to 0.3 mg/mL in 20 mM HEPES pH 7.5, 150 mM NaCl, 1 mM TCEP Buffer was incubated with compounds on ice for 45 min before being analysed. GCN2 stability was tested in the presence of 200 μM AZD1775 and 100 μM tRNA. A gradient of 0.5 °C/min was applied to the sample starting from 25 °C to 90 °C. Fluorescence at 330 nm and 350 nm was monitored using an excitation wavelength of 280 nm. Turbidity and scattering were also measured simultaneously. The intrinsic fluorescence of aromatic residues was measured at 330 and 350 nm. Data analysis and identification of inflection points/TMs were conducted using the Panta Analysis Software (NanoTemper). The first derivative was derived by the Panta software from which the $T_m$ were inferred. The thermal unfolding assays, concurrent treatment of AZD1775 and tRNA, were performed once (biological $n = 1$). However, the biphasic nature of GCN2 activation was reproducibly observed in more than a dozen independent thermal unfolding assays conducted in separate experiments.

### ADP-GLO assays

Kinase assays were conducted using an ADP-GLO Kinase assay kit (PROMEGA) in a 384 well plate. Two hundred nanomolar 200 nM GCN2 was mixed with a serial dilution AZD1775. To this, 250 μM ATP or 250 μM ATP and 10 μM eIF2α were added to reach a final kinase reaction of 2 μL in Reaction Buffer (Final Reaction Conditions 100 nM GCN2, 125 μM ATP, 5 μM eIF2α in 20 mM Tris pH 8.0, 150 mM NaCl, 0.1% BSA, 0.1% Tween 20, 2 mM TCEP). The reactions occurred for 40 min at RT and were quenched using the ADP-Glo reagent 1:1 volume ratio. ADP-GLO Reagent and Kinase Detection Reagent were then added as described in the ADP GLO Kinase Kit Protocol. Luminescence was measured using a Microplate reader (BMG PHERAstar FSX). All experiments were performed in quadruplet. The relative luminescence units were transformed to an ADP concentration by measuring a standard ATP/ADP curve as described by the ADP-GLO Kit Protocol. The data were analysed using Graphpad Prism 10, and IC50/EC50 values were calculated using a non-linear analysis, using the Inhibition or Biphasic inhibition functions.

### Reporting summary

Further information on research design is available in the Nature Portfolio Reporting Summary linked to this article.

## Data availability

The sequencing data generated in this study are available from the European Nucleotide Archive (ENA). The AZD1775 CRISPRi screen of the RPE-1 $TP53^{-/-}$ dCas9-KRAB cell line raw sequencing data are available under accession number PRJEB93993, and the AZD1775 ribosome profiling (Ribo-seq) raw sequencing data from the RPE-1 $TP53^{-/-}$ cell line are available under accession number PRJEB93983. Other data generated or analysed during this study are either included in this article, its supplementary materials, the source data file or available from the corresponding author upon request. Source data are provided with this paper.

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

## Acknowledgements

We thank members of the Jackson laboratory, as well as members of the Marciniak, Masson, Biffi, Villunger and Loizou laboratories for their assistance, discussion and reagents. We thank the Jacob Corn laboratory, ETH Zurich for their discussions and for providing the DDR focused CRISPRi library and RPE-1 dCas9-KRAB cell line. We are grateful to the Nathanael S. Grey laboratory, Stanford University for providing HRZ-1-057-1, HRZ-1-098-1 and ZNL-02-047 WEE1 degrader compounds. We also acknowledge the Research Instrumentation and Cell Services, Flow Cytometry and Microscopy Core Facilities at the CRUK Cambridge Institute for technical support and access to equipment and reagents. We thank the labs of Marcel Van Vugt (Groningen) and Daniel Durocher (Toronto) for their discussions and sharing of unpublished data. We also thank Anna Schrempf (Loizou/Winter labs, Vienna) and Gonçalo Oliveira (Loizou/Hanzlíková labs, Vienna/Prague) for their discussions and sharing of CRISPR screening protocols. We thank Manu Hegde and Eszter Zavodszky, LMB, Cambridge, for their insightful discussions and assistance with ribosomal assays. We thank Alice Dubois-Veltman of the Balasubramanian Lab, CRUK CI, for drawing chemical structures on ChemDraw for Fig. 5a. We also appreciate the informative discussions with members of the AstraZeneca Oncology R&D team in Cambridge. CRISPRi libraries were sequenced by Biomedical Sequencing Facility (BSF) (https://www.biomedical-sequencing.org). Ribo-seq libraries were generated, sequenced and analysed by EIRNABio (https://eirnabio.com). This research was supported by the European Research Council (ERC) under the European Union's Horizon 2020 research and innovation programme (Grant agreement No. 855741-DDREAMM-ERC-2019-SyG). Research in the S.P.J laboratory is supported by Cancer Research UK (CRUK) Discovery Award DRCPGM\100005, CRUK core grant C9545/A29580 and SEBINT-2024/100003 and ERC Synergy Award 855741 (DDREAMM); A.H. is funded by a grant to S.P.J from GSK; C.G.G. by an A*STAR National Science Scholarship; F.B.D. by a Harding Distinguished Postgraduate Scholarship; J.C.J.W. and S.L. by ERC Synergy Award 855741; and A.S.B. by Wellcome Early Career Award (227014/Z/23/Z). S.J.M. is supported by the MRC, MCMB MR/Y011813/1. J.Z. is funded by the Doctoral Training Programme in Medical Research (DTP-MR) and the Cambridge Trust. G.R.M. and V.V. (MRC iCase Studentship (MR/R01579/1) (BBSRC Capital Equipment Grant BB/V019635/1). J.E.C. is supported by the NOMIS Foundation, the Lotte and Adolf Hotz-Sprenger Stiftung, the Swiss National Science Foundation (project grants 310030_188858, 310030_201160 and 320030-227979), and the European Research Council (ERC) under the European Union's Horizon 2020 research and innovation programme (grant agreement No 855741, DDREAMM). The lab of J.E.C. has funded collaborations with Allogene, Cimeio, and Serac. J.F. is a recipient of the EMBO Postdoctoral Fellowship (ALTF 220-2021). M.F.S. is a recipient of the Boehringer Ingelheim Fonds (BIF) PhD fellowship.

## Author contributions

J.CJ.W. conceived the study, planned and performed most of the experiments, analysed data, and wrote the original draft. A.S.B. performed and analysed all immunofluorescence experiments. J.Z., with the supervision of S.J.M., generated lysates for phospho-GCN2 in vitro experiments and performed ATF4 reporter experiments. V.V. and G.R.M. performed ADP Glo assay in vitro experiments and thermal unfolding assays. E.G.L. with the supervision of G.B., cultured, treated, and generated lysates of PDAC patient-derived organoids. S.L. performed bioinformatic analyses of the CRISPRi screen. A.H., C.G.G., and F.B.D. assisted with western blots and RT-qPCR assays. H.R. provided unpublished WEE1 molecular glue compounds. L.K., Z.K., J.F., and M.F.S. with the supervision of J.E.C., designed and cloned the CRISPRi library. J.L. and A.V. supervised J.C.J.W. when performing the CRISPRi screen. A.S.B. and S.P.J. supervised J.C.J.W. for the remainder of the study. J.C.J.W., with assistance from A.S.B., generated the figures. A.S.B. helped with editing of the manuscript, with S.P.J. co-writing the manuscript.

## Competing interests

S.P.J. is a founding partner of Ahren Innovation Capital LLP, is co-founder of Mission Therapeutics Ltd., and Chief Research Officer (part-time) of Insmed Innovation UK. J.E.C. is a co-founder and SAB member of Serac Biosciences and an SAB member of Mission Therapeutics, Relation Therapeutics, Hornet Bio, and Kano Therapeutics. The remaining authors declare no competing interests.
