## [Transparent Peer review file · Nature Communications]

WEE1 inhibitors synergise with mRNA translation defects via activation of the kinase GCN2

Corresponding Author: Professor Stephen Jackson

Version 0:

Reviewer comments:

Reviewer #1

(Remarks to the Author)

This manuscript describes an off-target activity of AZD1775 and other WEE1 inhibitors. This off-target activity leads to activation of the GCN2 kinase and to decreased protein synthesis, and affects both normal and cancer cells. As such, it may contribute to the toxicity of WEE1 inhibitors and to the difficulty in developing these inhibitors in the clinic.

Specific Comments

1. I found the study to be quite interesting. Overall it is a well-performed, technically-solid study. In considering whether it should be published in Nature Communications, one could consider that a study describing that WEE1 inhibitors activate GCN2 was published last year in Cell Reports [Drainas AP, et al. GCN2 is a determinant of the response to WEE1 kinase inhibition in small-cell lung cancer. Cell Reports, 08 Aug 2024, 43(8):114606]. The Drainas et al. study, which the authors have cited, detracts from the novelty of the current manuscript.

However, on the positive side, the current manuscript is a solid study that provides additional information over what has been published. Moreover, validating the data of Drainas et al. is important for the field to quickly recognize that WEE1 inhibitors have an off-target activity that will affect their efficacy and toxicity in the clinic. One hopes that these findings will motivate the development of WEE1 inhibitors that exhibit greater cancer specificity.

2. WEE1 inhibitors are not the only kinase inhibitors that activate GCN2. It appears that several "highly specific" kinase inhibitors activate GCN2. Therefore, activation of GCN2 may need to become a standard assay to evaluate the specificity of kinase inhibitors. Perhaps, the authors can mention this in the Discussion.

3. The authors propose that the mechanism by which WEE1 inhibitors activate GCN2 involves binding of the WEE1 inhibitor to one subunit of a GCN2 dimer, which then leads to activation of the kinase activity of the second subunit of the GCN2 dimer. This is the model proposed by Drainas et al, who provided somewhat limited data to support it. It would have been nice to provide some data to support the proposed mechanism by which WEE1 inhibitors activate GCN2. I wonder if the authors could employ CETSA to determine if WEE1 inhibitors bind to GCN2; they could use endogenous GCN2 or ectopically expressed FLAG GCN2 (as shown in Fig. 4d) for this experiment. Neratinib and RP-3606 could serve as positive and negative controls, respectively. In the CETSA experiment, GCN2 levels could be determined by immunoblotting, rather than by mass spec, since the latter is less quantitative. If this experiment does not work, I would not ask the authors to pursue further studies to explore how the WEE1 inhibitor activates GCN2, as this effort could become open-ended.

4. It is helpful that the authors showed that other kinase inhibitors in the DDR field, eg ATR and PKMYT inhibitors etc do not activate GCN2, as this is a question that many readers will have.

5. For all graphs of the type shown in Figs 2b, 2c, 3f, 5c, etc, can the authors calculate EC50 values?

6. The WEE1 kinase inhibitor AZD1775 used in this study is also a potent PLK1 inhibitor (doi.org/10.1021/acs.jmedchem.7b00996). Since the authors are discussing off-target effects of AZD1775, they should also discuss how inhibition of PLK1 affects normal and cancer cells.

Reviewer #2

(Remarks to the Author)

In this study Wilson et al. identify a novel mechanism of action for WEE1 kinase inhibitors. In addition to their effect on inducing genotoxic stress by inhibiting WEE1 directly and thereby overriding the G2-M checkpoint, the authors find that WEE1 inhibitors also activate the Integrated Stress Response (ISR) through an off-target activation of the GCN2 kinase. Using a CRISPRi screen, the authors identify two mRNA translation-associated genes, GSPT1 and ALKBH8, whose depletion sensitizes cells to the WEE1 inhibitor AZD1775. The combination of WEE1 inhibitors and defects in translation machinery (e.g. via GSPT1 degradation using CC-90009) synergistically enhances cytotoxicity through GCN2-mediated ISR activation. Importantly, the authors show that this ISR activation is independent of WEE1 inhibition, suggesting an off-target effect of ATP-competitive WEE1 inhibitors. PROTACs targeting WEE1 do not activate the ISR, yet still retain genotoxic activity, offering a more selective and potentially safer therapeutic strategy.

Thus, the study proposes a dual-toxicity model for WEE1 inhibitors: an ISR-dependent toxicity (via an off-target GCN2 activation) and an ISR-independent genotoxicity (via on-target WEE1 inhibition). Moreover, this study emphasizes the therapeutic advantage of combining WEE1 inhibitors with translational stress-inducing agents, or switching to targeted degraders of WEE1 to mitigate side effects.

Together I think that this manuscript represents an important and timely contribution to the field of cancer therapeutics. The identification of the off-target ISR activation by WEE1 inhibitors via the GCN2 kinase is both novel and highly relevant to clinical oncology, especially as WEE1 inhibitors continue to progress through clinical trials. The demonstration that PROTAC degraders of WEE1 can mitigate ISR toxicity adds clear translational value.

However, while the manuscript is mechanistically and methodologically sound, I have the following concerns:

1. The absence of in vivo data

The entire dataset is generated using in vitro cell line experiments. Although ISR activation and drug synergy are validated in multiple cancer and non-cancer cell line models, I miss an in vivo validation of the findings, e.g. using PDX or genetically engineered mouse models. How relevant is the off-target effect in vivo compared to a pure (genetic or PROTAC-induced) WEE1 inactivation? Especially because this is a study with clear therapeutic implications, I think that such in vivo data would be important. Patient-derived organoid/tumoroid models would also be a step in this direction, in particular to assess the physiological relevance and translational potential of the ISR activation findings.

2. Confirming direct binding of WEE1 to GCN2

In lines 276-278, the authors state: "From this, we concluded that the off-target activity of AZD1775 and other WEE1 inhibitors was likely due to direct binding and activation of GCN2". The authors come to this conclusion by carrying out in vitro phosphorylation assays with GCN2 in the presence of ATP and various small-molecule inhibitors. While this is indeed useful and suggestive, I miss more evidence for the direct binding. What about alphaFold-3-based predictions of the interaction? Could one not mutate specific sites amino acids in the binding pocket that would then abrogate the interaction? Or do thermal shift assays in the presence of the drug to confirm the binding?

3. What about clinical correlations or biomarker stratification?

Given the ISR pathway's complexity and potential toxicity implications, the manuscript may benefit from additional correlative data using clinical data, e.g. ISR signatures in patient samples or sensitivity profiles. Any evidence from the clinical data to support the concept?

Minor Points:

4. Fig. 1: Did the CRISPRi screen include constructs targeting GCN2? If yes where they enriched after WEE1i selection?
5. The authors attribute ISR activation to GCN2 off-target binding of WEE1 inhibitors and cite a study with kinome profiling data (reference 51). This is currently mentioned in passing and could be expanded more and better explained.

Reviewer #3

(Remarks to the Author)

The manuscript by Wilson et al. suggests that clinically developed Wee1 inhibitors directly induce the GCN2 eIF2 kinase and integrated stress response (ISR), leading to toxicity that can be omitted by targeted depletion of Wee1 protein. This is a timely and significant line of investigation that supports the emerging publications suggesting that eIF2 kinases can be activated by many small molecules designed to be inhibitors of protein kinases. The manuscript is clearly written, flows logically and is succinct. The manuscript data largely supports the stated conclusions. There are some concerns, involving the need for an additional control and inclusion of more quantitation and statistical analyses. The apparent requirement for GCN1 in the model for AZD1775 activation of GCN2 is also not fully clear. Addressing these concerns would enhance the overall rigor and bolster the stated conclusions.

Reviewer concerns:

1. Data presentation and analyses: Statistics are missing from many of the experiments in the manuscript. See Fig. 1C, 1E, 1F, 3A, S5b as examples. It would be helpful to include the full quantitation and statistical support in the figures and methods and figure legends. Include some significance symbols in the figure panels. Many immunoblot panels are overly cropped,

lack molecular weight markers. Key immunoblot panels should include quantitation and statistical analyses. Is there a scale bar in Fig. 1D?

2. The manuscript suggests that the Wee1 inhibitor AZD1775 binds directly to GCN2 and activates the ISR and the manuscript provides evidence supporting this claim. For example, kinome profiling shows that AZD1775 binds to the GCN2 kinase domain, which is perhaps not surprising given the similarities between Wee1 and GCN2 kinase domains. A distinct feature of the proposed mechanism of direct activation of GCN2 by Wee1 inhibitors is a requirement for the accessory protein GCN1; in contrast to other GCN2 activators suggested in the literature. As a control, it would be expected that the GCN1 would not be required by ER stressed induced PERK, which would show that the ISR is intact and help validate the utility of the GCN1 KO (extended from Fig. S5b). What would be the the role of GCN1 in this process for a drug that is suggested to directly engage the GCN2 protein kinase domain?

3. Of interest, a PROTAC version of AZD1775 does not activate GCN2 and the ISR to an appreciable extent. Does the PROTAC version of AZD1775 block the interaction with GCN2?

4. The authors refer to “off-target ISR toxicity” by Wee1 inhibitors. Is the reader to infer that induction of the ISR contributes to the toxicity of Wee1 inhibitors, the efficacy of Wee1 inhibitors, or both? Are the authors proposing that a PROTAC approach to Wee1 inhibition, which appears to prevent appreciable ISR activation, will be more efficacious?

5. Data showing immunofluorescent staining of nuclear ATF4 is not shown in Fig. S8b for cell lines expressing guides targeting CDK2, cyclin A2, cyclin E1, cyclin E2, or AAVS1.

Version 1:

Reviewer comments:

Reviewer #1

(Remarks to the Author)

The authors have addressed my comments, which were minor. The manuscript is suitable for publication.

Reviewer #2

(Remarks to the Author)

I am satisfied with the answers on my comments. Thank you for your efforts.

Reviewer #3

(Remarks to the Author)

The revised manuscript reports that clinically developed Wee1 inhibitors directly induce the GCN2 eIF2 kinase and integrated stress response (ISR), leading to toxicity that can be omitted by targeted depletion of Wee1 protein. This is a timely and significant line of investigation that supports the emerging publications suggesting that eIF2 kinases can be activated by many small molecules designed to be inhibitors of protein kinases. The manuscript is clearly written, flows logically and is succinct. The manuscript data supports the stated conclusions. Prior concerns were addressed. This manuscript provides important new insights into how activation of the ISR can be a complication for many therapeutic strategies and how this complication can be overcome.

REVIEWER COMMENTS

Reviewer #1 (Remarks to the Author):

This manuscript describes an off-target activity of AZD1775 and other WEE1 inhibitors. This off-target activity leads to activation of the GCN2 kinase and to decreased protein synthesis, and affects both normal and cancer cells. As such, it may contribute to the toxicity of WEE1 inhibitors and to the difficulty in developing these inhibitors in the clinic.

Specific Comments

1. I found the study to be quite interesting. Overall it is a well-performed, technically-solid study. In considering whether it should be published in Nature Communications, one could consider that a study describing that WEE1 inhibitors activate GCN2 was published last year in Cell Reports [Drainas AP, et al. GCN2 is a determinant of the response to WEE1 kinase inhibition in small-cell lung cancer. Cell Reports, 08 Aug 2024, 43(8):114606]. The Drainas et al. study, which the authors have cited, detracts from the novelty of the current manuscript.

However, on the positive side, the current manuscript is a solid study that provides additional information over what has been published. Moreover, validating the data of Drainas et al. is important for the field to quickly recognize that WEE1 inhibitors have an off-target activity that will affect their efficacy and toxicity in the clinic. One hopes that these findings will motivate the development of WEE1 inhibitors that exhibit greater cancer specificity.

We thank the Reviewer 1 for their thoughtful and positive evaluation of our work. We acknowledge the contribution of the Drainas et al. study in identifying GCN2 activation as a component of the cellular response to WEE1 inhibition in lung cancer cell lines. We believe that our study builds upon these foundational findings and extends them in several important ways:

Expanded model systems and clinical relevance: We test WEE1 inhibitor-induced ISR activation beyond SCLC cell lines, demonstrating this phenomenon across diverse cancer and non-cancer human and mouse cell lines, as well as patient-derived organoid models, thereby establishing broader biological relevance.

Further characterisation of WEE1i-induced ISR: We provide additional mechanistic insights, including cell cycle analysis and demonstration of the bell-shaped/paradoxical dose-response curve of GCN2 activation with AZD1775. We also evaluate multiple clinically relevant WEE1 inhibitors that include the Debio0123 and Zn-c3 compounds, expanding the therapeutic implications.

Newly described synergistic interactions and drug combinations: We identify new synergistic relationships between WEE1 inhibitors and GSPT1 degradation/depletion as well as ALKBH8 depletion, while demonstrating that WEE1 inhibitor synergies involving nucleotide metabolism and mitotic defects remain largely independent of ISR activation.

Comparative analysis with alternative approaches: We conduct comparisons between WEE1 inhibitors and other DDR compounds, WEE1 PROTACs, and recently developed WEE1 molecular glues, providing insights into on-target versus off-target effects.

Demonstration of GCN1 dependency: We demonstrate that GCN2 activators, including WEE1 inhibitors, GCN2iB, and neratinib, require GCN2's interactor GCN1 for cellular ISR signalling - a previously underappreciated aspect of ISR regulation that furthers our understanding of GCN2 activators.

2. WEE1 inhibitors are not the only kinase inhibitors that activate GCN2. It appears that several "highly specific" kinase inhibitors activate GCN2. Therefore, activation of GCN2 may need to become a standard assay to evaluate the specificity of kinase inhibitors. Perhaps, the authors can mention this in the Discussion.

We agree. Given the promiscuous nature of GCN2, it is very likely that future kinase inhibitors developed will also have the ability to activate GCN2 at relevant concentrations. Accordingly, we have included an extra paragraph in the discussion (page 16 in manuscript):

“Recent publications have demonstrated that WEE1 inhibitors are not unique in their ability to activate the GCN2 kinase. A plethora of seemingly specific kinase inhibitors can trigger GCN2 activation, highlighting a broader concern regarding the off-target engagement of this stress-response pathway. As such, activation of GCN2 may need to become a standard assay to evaluate the specificity of kinase inhibitors. We encourage future studies, and perhaps drug development pipelines, to incorporate GCN2 activation assays as part of a routine specificity screen.”

3. The authors propose that the mechanism by which WEE1 inhibitors activate GCN2 involves binding of the WEE1 inhibitor to one subunit of a GCN2 dimer, which then leads to activation of the kinase activity of the second subunit of the GCN2 dimer. This is the model proposed by Drainas et al, who provided somewhat limited data to support it. It would have been nice to provide some data to support the proposed mechanism by which WEE1 inhibitors activate GCN2. I wonder if the authors could employ CETSA to determine if WEE1 inhibitors bind to GCN2; they could use endogenous GCN2 or ectopically expressed FLAG GCN2 (as shown in Fig. 4d) for this experiment. Neratinib and RP-3606 could serve as positive and negative controls, respectively. In the CETSA experiment, GCN2 levels could be determined by immunoblotting, rather than by mass spec, since the latter is less quantitative. If this experiment does not work, I would not ask the authors to pursue further studies to explore how the WEE1 inhibitor activates GCN2, as this effort could become open-ended.

CETSA assays were performed on HEK293 WT cells using the protocol described by Jafari et al. (Nat Protoc 9, 2100–2122, 2014; <https://doi.org/10.1038/nprot.2014.138>). In our first test, we probed both endogenous GCN2 and WEE1 following a 30µM 1-hour AZD1775 treatment. From our initial experiment, we observed that AZD1775 induced a strong thermal stabilisation phenotype on WEE1 from 40°C to 52°C.

We initially noted a possible change in GCN2 stability at 56 °C and 60 °C in response to AZD1775 treatment, prompting further investigation. However, in subsequent experiments, any apparent changes in GCN2 stability or its interaction with known binding partners were not reproducible. See below, a replicate that includes different GCN2 activators.

Of note, a previously published study using *in vitro* thermal stability assays (Szaruga et al., *Nat Commun* 14, 5535, 2023; <https://doi.org/10.1038/s41467-023-40823-8>, supplementary fig. 9) reported rather modest melting temperature shifts upon binding of different GCN2 interactors (100 μM) to the GCN2 kinase domain. These compounds included 100 μM GSK157 (2.1°C shift), 100 μM GSK413 (1.4°C shift), 100 μM AMG44 (3°C shift) and 100 μM C16 (4.1°C shift).

Taken together, these findings suggest that CETSA, when coupled with western blot detection, can detect the stabilisation phenotype shift of WEE1 upon AZD1775 binding. However, it lacks the sensitivity required to reliably detect the subtle changes in GCN2 melting temperature that occur upon ligand binding.

We have additional GCN2-AZD1775 data from *in vitro* ADP-Glo and *in vitro* thermal unfolding assays. Please see reviewer 2, point number 2 for further information on this.

4. It is helpful that the authors showed that other kinase inhibitors in the DDR field, eg ATR and PKMYT inhibitors etc do not activate GCN2, as this is a question that many readers will have.

We thought that it was particularly important to confirm that the PKMYT1 inhibitor RP-6306 does not activate GCN2, given that it inhibits a WEE1 family member and is currently being evaluated in clinical trials. We tested RP-6306 both in combination with WEE1 inhibitors (a combination currently tested in clinical trials) and in our *in vitro* GCN2 autophosphorylation assays, where it demonstrated lower potency than WEE1 inhibitors at inducing GCN2 activation. We believe that this will be of interest to readers who are specialised in the DDR field.

5. For all graphs of the type shown in Figs 2b, 2c, 3f, 5c, etc, can the authors calculate EC50 values?

EC50 values have been added to all resazurin viability assay graphs, including those listed by the reviewer.

6. The WEE1 kinase inhibitor AZD1775 used in this study is also a potent PLK1 inhibitor ([//doi.org/10.1021/acs.jmedchem.7b00996](https://doi.org/10.1021/acs.jmedchem.7b00996)). Since the authors are discussing off-target effects of AZD1775, they should also discuss how inhibition of PLK1 affects normal and cancer cells.

We cited this informative study when we briefly discussed the study's *in vitro* kinome profiling, but we had only mentioned that PLK1 was a top 'hit' in the kinome screen without discussing their validation experiments. We have expanded on the main text, where we mentioned the kinome scan. We have now expanded the main text (page 11 and 12 of the manuscript) to include their dual WEE1-PLK1 validation findings, where they demonstrated both phospho-CDK1 (WEE1 target) and phospho-TCTP (PLK1 target) induction upon AZD1775 treatment across both cancer and non-cancer cell lines.

“Notably, a previous in vitro kinome profile study, which systematically evaluated the binding affinity of compounds across the kinome in vitro (403 wild-type and 65 mutant kinases), had shown that 0.5 μM AZD1775 was highly selective to the second domain of GCN2 along with WEE1, WEE2, PLK1, among others (Supplementary Fig.14c). The study validated dual WEE1-PLK1 inhibition by observing simultaneously attenuated levels of phosphorylated CDK1 (Tyr-15), a canonical WEE1 target, and TCTP (Ser-46), a previously described phosphorylation site of PLK1, in synchronised noncancer and cancer cell lines upon AZD1775 treatment. Taken together, these findings underscore that AZD1775 can modulate multiple kinase pathways in parallel.”

Reviewer #2 (Remarks to the Author):

In this study Wilson et al. identify a novel mechanism of action for WEE1 kinase inhibitors. In addition to their effect on inducing genotoxic stress by inhibiting WEE1 directly and thereby overriding the G2-M checkpoint, the authors find that WEE1 inhibitors also activate the Integrated Stress Response (ISR) through an off-target activation of the GCN2 kinase.

Using a CRISPRi screen, the authors identify two mRNA translation-associated genes, GSPT1 and ALKBH8, whose depletion sensitizes cells to the WEE1 inhibitor AZD1775. The combination of WEE1 inhibitors and defects in translation machinery

(e.g. via GSPT1 degradation using CC-90009) synergistically enhances cytotoxicity through GCN2-mediated ISR activation. Importantly, the authors show that this ISR activation is independent of WEE1 inhibition, suggesting an off-target effect of ATP-competitive WEE1 inhibitors. PROTACs targeting WEE1 do not activate the ISR, yet still retain genotoxic activity, offering a more selective and potentially safer therapeutic strategy.

Thus, the study proposes a dual-toxicity model for WEE1 inhibitors: an ISR-dependent toxicity (via an off-target GCN2 activation) and an ISR-independent genotoxicity (via on-target WEE1 inhibition). Moreover, this study emphasizes the therapeutic advantage of combining WEE1 inhibitors with translational stress-inducing agents, or switching to targeted degraders of WEE1 to mitigate side effects.

Together I think that this manuscript represents an important and timely contribution to the field of cancer therapeutics. The identification of the off-target ISR activation by WEE1 inhibitors via the GCN2 kinase is both novel and highly relevant to clinical oncology, especially as WEE1 inhibitors continue to progress through clinical trials. The demonstration that PROTAC degraders of WEE1 can mitigate ISR toxicity adds clear translational value.

However, while the manuscript is mechanistically and methodologically sound, I have the following concerns:

1. The absence of *in vivo* data

The entire dataset is generated using *in vitro* cell line experiments. Although ISR activation and drug synergy are validated in multiple cancer and non-cancer cell line models, I miss an *in vivo* validation of the findings, e.g. using PDX or genetically engineered mouse models. How relevant is the off-target effect *in vivo* compared to a pure (genetic or PROTAC-induced) WEE1 inactivation? Especially because this is a study with clear therapeutic implications, I think that such *in vivo* data would be important. Patient-derived organoid/tumoroid models would also be a step in this direction, in particular to assess the physiological relevance and translational potential of the ISR activation findings.

We thank Reviewer 2 for their detailed and important suggestions. We appreciate the reviewer's suggestion regarding *in vivo* validation, which would provide valuable translational insights. We emphasise that the primary scope of this study was to elucidate the mechanistic basis of ISR activation following WEE1 inhibitor treatment using well-characterised cell line models and *in vitro* systems, which allowed us to perform a comprehensive molecular mechanism.

Recognising the importance of translational relevance, we have now included patient-derived organoid (PDO) of pancreatic cancer origin experiments to bridge our mechanistic findings with clinically relevant models. As shown in Figure 2e,f of the updated manuscript, we demonstrate that ATF4 induction upon 500 nM AZD1775 treatment and its rescue by 500 nM ISRIB co-treatment in pancreatic cancer PDOs. AZD1775 has previously been evaluated in pancreatic cancer clinical trials (e.g.

NCT02037230, NCT02194829), highlighting its clinical relevance. The western blots below have been added to Fig.2e and f of the manuscript.

Human organoids hM1a (annotated as PDO #1) and hF24 (annotated as PDO #2) have been previously published (Tiriác, H. *et al.* Organoid Profiling Identifies Common Responders to Chemotherapy in Pancreatic Cancer. *Cancer Discov.* **8**, 1112–1129 (2018)).

Importantly, both WEE1 inhibitors and ISRIB (unlike GCN2iB) are currently being evaluated in clinical trials, providing immediate translational potential for a combination therapy approach. In contrast, while WEE1 molecular glues and PROTACs have been valuable tools for mechanistic studies in cell culture, to our knowledge these recently developed compounds have not been tested *in vivo* and their bioavailability, and other pharmacokinetic parameters remain unknown, limiting their immediate clinical applicability. We have included an extra paragraph (page 15 and 16 of the manuscript) in our discussion to address this:

*“While WEE1 molecular glues and PROTACs represent valuable tools for mechanistic studies, these compounds were primarily developed for cell culture applications and their *in vivo* pharmacokinetic parameters remain unclear. In contrast, an immediate translational approach to minimise the ISR activation from WEE1 inhibitor treatment may be to combine WEE1 inhibitors with ISRIB, as ISRIB has more established *in vivo* toxicology profiles and has exhibited minimal overt toxicity at physiological concentrations where it demonstrated efficacy. To assess the translational potential of this combination strategy, we tested WEE1 inhibitors combined with ISRIB in pancreatic patient-derived organoids, demonstrating that this approach can mitigate ISR activation whilst preserving DNA damage induction in clinically relevant models.”*

The mechanistic insights provided, validated across multiple cell line models and supported by PDO data, establish a foundation for future *in vivo* investigations. We believe that the current study makes a significant contribution by defining the molecular mechanism of ISR activation by WEE1 inhibitors while the PDO experiments provide additional evidence for translational relevance.

2. Confirming direct binding of WEE1 to GCN2. In lines 276-278, the authors state: “From this, we concluded that the off-target activity of AZD1775 and other WEE1

inhibitors was likely due to direct binding and activation of GCN2". The authors come to this conclusion by carrying out *in vitro* phosphorylation assays with GCN2 in the presence of ATP and various small-molecule inhibitors. While this is indeed useful and suggestive, I miss more evidence for the direct binding. What about alphaFold-3-based predictions of the interaction? Could one not mutate specific sites amino acids in the binding pocket that would then abrogate the interaction? Or do thermal shift assays in the presence of the drug to confirm the binding?

To further test whether AZD1775 was capable of directly binding and activating GCN2, we used recombinantly expressed and purified human GCN2 in *in vitro* kinase assays and drug binding measurements. By monitoring ADP production of GCN2 in the presence of ATP and full-length human eIF2 α substrate, we observed the characteristic "bell shaped" curve of paradoxical activation (see fig 6 Tang CP et al., GCN2 kinase activation by ATP-competitive kinase inhibitors. *Nat Chem Biol.* 2022 doi: 10.1038/s41589-021-00947-8 or fig 9 Szaruga, M., et al. Activation of the integrated stress response by inhibitors of its kinases. *Nat Commun* <https://doi.org/10.1038/s41467-023-40823-8> for examples of the bell-shaped curve of GCN2 activation). Activation of GCN2 was observed to peak at 36 nM AZD1775, with a drop to baseline activity at lower concentrations ($EC_{50} = 16$ nM) and an inhibitory effect at higher concentrations ($IC_{50} = 89$ nM) with a complete ablation of kinase activity at ~ 10 μ M.

Using *in vitro* thermal unfolding assays, we observed biphasic thermal behaviour of GCN2 with both negative and positive first derivative F350/F330 peaks. A biphasic profile was also observed with GCN2 in the presence of its physiological activator tRNA, showing a -2.1°C shift in the negative peak (indicative of structural changes resulting in a reduction of tryptophan solvent exposure) and only a subtle -0.2°C shift in the positive thermal unfolding peak (indicative of exposed tryptophan residues). In contrast, AZD1775 addition eliminated the biphasic thermal profile of GCN2 and instead caused a more substantial -2.2°C shift in the unfolding transition peak.

This ADP-Glo (left) and thermal unfolding assays (right) have been added to Fig.4f and g of the manuscript.

3. What about clinical correlations or biomarker stratification?

Given the ISR pathway's complexity and potential toxicity implications, the manuscript may benefit from additional correlative data using clinical data, e.g. ISR signatures in

patient samples or sensitivity profiles. Any evidence from the clinical data to support the concept?

The reviewer raises an interesting point regarding clinical correlations with respect to the ISR. Unfortunately, such data are very much lacking in the field. The limited data that are available do suggest that ISR activation in particular cancers can lead to poorer survival outcomes. An example of such is a Nature Communications paper (<https://doi.org/10.1038/s41467-021-24661-0>) where p-eIF2 α levels were assessed via IHC in human lung adenocarcinoma samples after surgery (Fig. 1B), showing an association with poorer survival outcomes with high p-eIF2 α levels. Notably, WEE1 inhibitors have been previously evaluated in lung cancer clinical trials (e.g. NCT02593019). That said, we are aware of collaborators who have attempted IHC in *in vivo* tissue before to probe for p-eIF2 α and encountered significant issues with non-specific antibody binding. Also given the highly transient nature of p-eIF2 α , it is easy to imagine that tissue handling and fixation could disrupt the phosphorylation status, further complicating interpretation.

We note that the ISR related transcription factor ATF4 has been correlated with poorer survival outcomes in liver cancer (<https://www.proteinatlas.org/ENSG00000128272-ATF4/cancer>) and the ISR related transcription factor DDIT3/CHOP has been correlated with poorer survival outcomes in liver cancer, melanoma and renal cancer (<https://www.proteinatlas.org/ENSG00000175197-DDIT3/cancer>). However, the DDIT3 antibody used has an uncertain reliability score in the human protein atlas (i.e. a low consistency between antibody staining and the RNA expression data recorded).

Taken together, the limited data that are publicly available suggest that ISR activation in particular cancer types may lead to poorer survival outcomes. This supports a hypothesis that WEE1 inhibitor strategies that limit ISR activation versus traditional WEE1 inhibitor treatment alone could yield better survival outcomes to particular cancers.

Minor Points:

4. Fig. 1: Did the CRISPRi screen include constructs targeting GCN2? If yes where they enriched after WEE1i selection?

sgRNAs targeting GCN2 were not in our original CRISPRi library.

5. The authors attribute ISR activation to GCN2 off-target binding of WEE1 inhibitors and cite a study with kinome profiling data (reference 51). This is currently mentioned in passing and could be expanded more and better explained.

The text mentioning the kinome profiling has been expanded. Thus, we have included that this evaluated the *in vitro* binding of 403 wild-type and 65 mutant kinases and we have included the concentration of AZD1775 used (0.5 μ M). The authors of this study also validated the PLK1 inhibition from AZD1775 treatment in cell lines by testing the PLK1 phosphorylation target p-TCTP (Ser-46) which we have also included (page 11 and 12 of manuscript).

“Notably, a previous in vitro kinome profile study, which systematically evaluated the binding affinity of compounds across the kinome in vitro (403 wild-type and 65 mutant kinases), had shown that 0.5 μM AZD1775 was highly selective to the second domain of GCN2 along with WEE1, WEE2, PLK1, among others (Supplementary Fig.14c). The study validated dual WEE1-PLK1 inhibition by observing simultaneously attenuated levels of phosphorylated CDK1 (Tyr-15), a canonical WEE1 target, and TCTP (Ser-46), a previously described phosphorylation site of PLK1, in synchronised noncancer and cancer cell lines upon AZD1775 treatment. Taken together, these findings underscore that AZD1775 can modulate multiple kinase pathways in parallel.”

Reviewer #3 (Remarks to the Author):

The manuscript by Wilson et al. suggests that clinically developed Wee1 inhibitors directly induce the GCN2 eIF2 kinase and integrated stress response (ISR), leading to toxicity that can be omitted by targeted depletion of Wee1 protein. This is a timely and significant line of investigation that supports the emerging publications suggesting that eIF2 kinases can be activated by many small molecules designed to be inhibitors of protein kinases. The manuscript is clearly written, flows logically and is succinct. The manuscript data largely supports the stated conclusions. There are some concerns, involving the need for an additional control and inclusion of more quantitation and statistical analyses. The apparent requirement for GCN1 in the model for AZD1775 activation of GCN2 is also not fully clear. Addressing these concerns would enhance the overall rigor and bolster the stated conclusions.

We thank Reviewer 3 for their detailed and incisive comments and suggestions.

Reviewer concerns:

1. Data presentation and analyses: Statistics are missing from many of the experiments in the manuscript. See Fig. 1C, 1E, 1F, 3A, S5b as examples. It would be helpful to include the full quantitation and statistical support in the figures and methods and figure legends. Include some significance symbols in the figure panels. Many immunoblot panels are overly cropped, lack molecular weight markers. Key immunoblot panels should include quantitation and statistical analyses. Is there a scale bar in Fig. 1D?

- Statistical analysis has been added to the 6 well plate assays of Fig.1c and Fig.1f. Statistical analyses of 3 biological replicates of western blot Fig.1e were the fold change of GSPT1, ATF4 and puromycin were calculated. This has been added to Supplementary Fig.17a-c. Statistical analyses of western blot of 3 biological replicates of Fig 4c were the fold change of p-GCN2/GCN2, p-eIF2α/eIF2α and WEE1/vinculin were calculated. This has been added to Supplementary Fig.17d-f. Statistical analyses of 3 biological replicate western blots of the 650 nM AZD1775 or ZNL-02-096 treatments of Fig.5e were the fold change of ATF4, γH2AX and WEE1 were calculated. This adds further information on the comparison between AZD1775 and ZNL-02-096 and the dual phenomena that are elicited (genotoxicity and ISR activation) This has been added to Fig. 5f. Statistical analyses has been added to the ATF4 reporter

experiment in Supplementary Fig.5b. The p significance symbols from the quantitation have been added to all the respective figure legends.

- Molecular weight markers have been added to all western blots. Band cropping was readjusted when necessary to allow for the inclusion of an adjacent ladder marking.
- EC₅₀ value statistics have been added to all resazurin assays
- Scale bars have been added to all immunofluorescence representative images (Fig. 2d, Fig.4i and Supplementary Fig.8c).

2. The manuscript suggests that the Wee1 inhibitor AZD1775 binds directly to GCN2 and activates the ISR and the manuscript provides evidence supporting this claim. For example, kinome profiling shows that AZD1775 binds to the GCN2 kinase domain, which is perhaps not surprising given the similarities between Wee1 and GCN2 kinase domains. A distinct feature of the proposed mechanism of direct activation of GCN2 by Wee1 inhibitors is a requirement for the accessory protein GCN1; in contrast to other GCN2 activators suggested in the literature. As a control, it would be expected that the GCN1 would not be required by ER stressed induced PERK, which would show that the ISR is intact and help validate the utility of the GCN1 KO (extended from Fig. S5b). What would be the role of GCN1 in this process for a drug that is suggested to directly engage the GCN2 protein kinase domain?

The reviewer raises important points regarding the requirement of GCN1 for cellular activation of GCN2 following AZD1775 treatment. In response, we tested the reviewer's suggested control, the PERK activator (via ER stress) thapsigargin, as well as the GCN2 activator neratinib. While the reviewer mentions that a requirement for GCN1 appears to contrast with reports on other GCN2 activators, we point out that GCN1L1/GCN1 was among the top resistance hits in response to neratinib in published CRISPR screens (Fig. 1A, Fig. 2B, 2C, and Extended Fig. 1B in Tang et al., *Nat Chem Biol* 18, 207–215, 2022; <https://doi.org/10.1038/s41589-021-00947-8>). Although this observation was noted briefly in this study, it was not explored further experimentally by the authors.

In our own experiments, we observed that ATF4 induction following 6 hr thapsigargin treatment was comparable in both wild-type and GCN1 knockout HEK293 cells, consistent with the literature that thapsigargin activates the ISR via PERK rather than GCN2. In contrast, ATF4 levels were markedly reduced in GCN1 knockout cells treated with AZD1775, consistent with our previous findings. Similarly, neratinib-induced ATF4 expression was also reduced in the absence of GCN1, suggesting that neratinib, like AZD1775, may require GCN1 for effective ISR signalling.

To further validate these observations, we conducted resazurin assays in RPE-1 TP53^{-/-} dCas9-KRAB cells, where GCN1 depletion had previously conferred resistance to AZD1775 (Fig. 2c of manuscript). We found that GCN1 depletion did not confer resistance to 72-hour thapsigargin treatment but did result in a strong resistance phenotype upon 72-hour neratinib treatment with EC₅₀ values of the cell line expressing sgRNAs targeting the AAVS1 locus being 129 nM and two separate sgRNAs targeting GCN1 being 414 nM and 662 nM respectively. The western blot above and resazurin assays below have been incorporated into Supplementary Fig.5 of the manuscript.

Taken together, these data demonstrate that thapsigargin-induced ISR activation is independent of GCN1, while both AZD1775 and neratinib appear to rely on GCN1 for their ISR activity. Therefore, AZD1775 appears to be not unique in being a GCN1 dependent, GCN2 activator. Although the precise mechanistic role of GCN1 is beyond the scope of this study and would require in depth structural biology work, one leading hypothesis that we have is that GCN1 influences the conformation of GCN2 in a way that affects the accessibility of the ATP-binding pocket to certain ligands.

3. Of interest, a PROTAC version of AZD1775 does not activate GCN2 and the ISR to an appreciable extent. Does the PROTAC version of AZD1775 block the interaction with GCN2?

Given that WEE1 inhibitor constructs interact with GCN2 in cis to trigger ISR activation, and that the PROTAC version does not induce ISR to the same extent, we were left with two main hypotheses: either the PROTAC is degrading GCN2, or, as the reviewer mentioned, the PROTAC has a reduced capacity to interact with GCN2.

We note that in our western blots (Fig. 5e, Supplementary Fig.16c), we did not observe any visible degradation of GCN2 following PROTAC treatment. Indeed, previous mass spec data performed (Zhengnian Li et al., Development and Characterization of a Wee1 kinase Degrader, Cell Chemical Biology, <https://doi.org/10.1016/j.chembiol.2019.10.013>.) of the WEE1 PROTAC/ZNL-02-096, shows significant degradation of WEE1 and no significant degradation of GCN2. We've highlighted WEE1, GCN2 and ZFP91 (ZFP91 is an established off-target of pomalidomide-based degraders (<https://doi.org/10.1038/ncomms15398>)) in the dataset (see below).

Dataset from <https://doi.org/10.1016/j.chembiol.2019.10.013>.
-MOLT4 cells treated with ZNL-02-096 (100 nM) for 2 hours

This leads us to believe that AZD1775 being conjugated to the CRBN-binding ligand, pomalidomide via a linker reduces the GCN2 activation capacity of AZD1775 in a way that is independent of proteasomal degradation.

To test this, we performed further *in vitro* FLAG tagged GCN2 activation assays, this time to compare AZD1775 to the ZNL-02-096 WEE1 PROTAC at equimolar concentrations. We observed, from 3 separate biological experiments, that AZD1775 induced GCN2 phosphorylation at lower concentrations compared to ZNL-02-096. We included one experiment to Fig.5d and another biological experiment to Supplementary Fig.16b. From these experiments, we concluded that the PROTAC version of AZD1775 has a reduced capacity to activate GCN2 in cis.

Added to Fig.5d

Added to Supplementary Fig.16b

4. The authors refer to “off-target ISR toxicity” by Wee1 inhibitors. Is the reader to infer that induction of the ISR contributes to the toxicity of Wee1 inhibitors, the efficacy of Wee1 inhibitors, or both? Are the authors proposing that a PROTAC approach to Wee1 inhibition, which appears to prevent appreciable ISR activation, will be more efficacious?

We included off-target toxicity to be defined as a loss of cellular proliferation via a mechanism that is independent of WEE1 itself (off-target if we are to define WEE1 as the intended target). From this, we would say that the ISR contributes to the toxicity of WEE1 inhibitors in an off-target manner, which we think would resonate for readers specialised in the DNA damage response who are interested in purely eliciting replication stress from WEE1 inhibitors. If there are contexts in which it is desirable to both inhibit WEE1 and activate the ISR, or to drive a synergistic interaction via the likes of GSPT1 degradation, then we can also assert that the ISR contributes to efficacy.

We report that the PROTAC elicits more WEE1 mediated toxicity and less ISR activation overall. If we define WEE1 to be the intended target and WEE1 inhibition – and its subsequent genotoxicity to be the intended toxicity – then we can conclude from our data that the ZNL-02-096 PROTAC is more efficacious than AZD1775.

5. Data showing immunofluorescent staining of nuclear ATF4 is not shown in Fig. S8b for cell lines expressing guides targeting CDK2, cyclin A2, cyclin E1, cyclin E2, or AAVS1.

Representative images have been added to Supplementary Fig.8 with scale bar included.

Response to reviewers

We thank all three of the reviewers for their careful evaluation of our manuscript and for the constructive feedback provided. We are grateful for the improvements suggested in the first round of reviews.

Reviewer #1 (Remarks to the Author):

The authors have addressed my comments, which were minor. The manuscript is suitable for publication.

We thank the reviewer for their positive assessment and for concluding that the manuscript is now suitable for publication.

Reviewer #2 (Remarks to the Author):

I am satisfied with the answers on my comments. Thank you for your efforts.

We appreciate the reviewer's recognition of our efforts to address their comments.

Reviewer #3 (Remarks to the Author):

The revised manuscript reports that clinically developed Wee1 inhibitors directly induce the GCN2 eIF2 kinase and integrated stress response (ISR), leading to toxicity that can be omitted by targeted depletion of Wee1 protein. This is a timely and significant line of investigation that supports the emerging publications suggesting that eIF2 kinases can be activated by many small molecules designed to be inhibitors of protein kinases. The manuscript is clearly written, flows logically and is succinct. The manuscript data supports the stated conclusions. Prior concerns were addressed. This manuscript provides important new insights into how activation of the ISR can be a complication for many therapeutic strategies and how this complication can be overcome.

We thank the reviewer for their summary of our work and for recognising that the revised manuscript addresses prior concerns. We are appreciative of the reviewer's acknowledgment that our study provides important new insights into ISR activation as a complication in therapeutic strategies and how it may be overcome. We thank the reviewer as well particularly for the feedback on further controls for the GCN1 perturbation cell lines which we think adds robustness to our stated conclusions.